# INTERLEAVING REASONING
# FOR BETTER TEXT-TO-IMAGE GENERATION

**Wenxuan Huang**[1,2]**, Shuang Chen**[4]**, Zheyong Xie**[3]**, Shaosheng Cao**[3✉]** , Shixiang Tang**[2]**,
Yufan Shen**[5]**, Qingyu Yin**[5]**, Wenbo Hu**[4]**, Xiaoman Wang**[1]**, Yuntian Tang**[1]**, Junbo Qiao**[1]**,
Yue Guo**[4]**, Yao Hu**[3]**, Zhenfei Yin**[6✉]** , Philip Torr**[6]**, Yu Cheng**[2]**, Wanli Ouyang**[2]**, Shaohui Lin**[1,7✉]

[1]East China Normal University     [2]The Chinese University of Hong Kong     [3]Xiaohongshu Inc.
[4]University of California, Los Angeles     [5]Zhejiang University     [6]University of Oxford
[7]Key Laboratory of Advanced Theory and Application in Statistics and Data Science - MOE, China
wxhuang@gmail.com (Wenxuan Huang)
shaohuilin007@gmail.com (Shaohui Lin)
Github Repo: https://github.com/Osilly/Interleaving-Reasoning-Generation

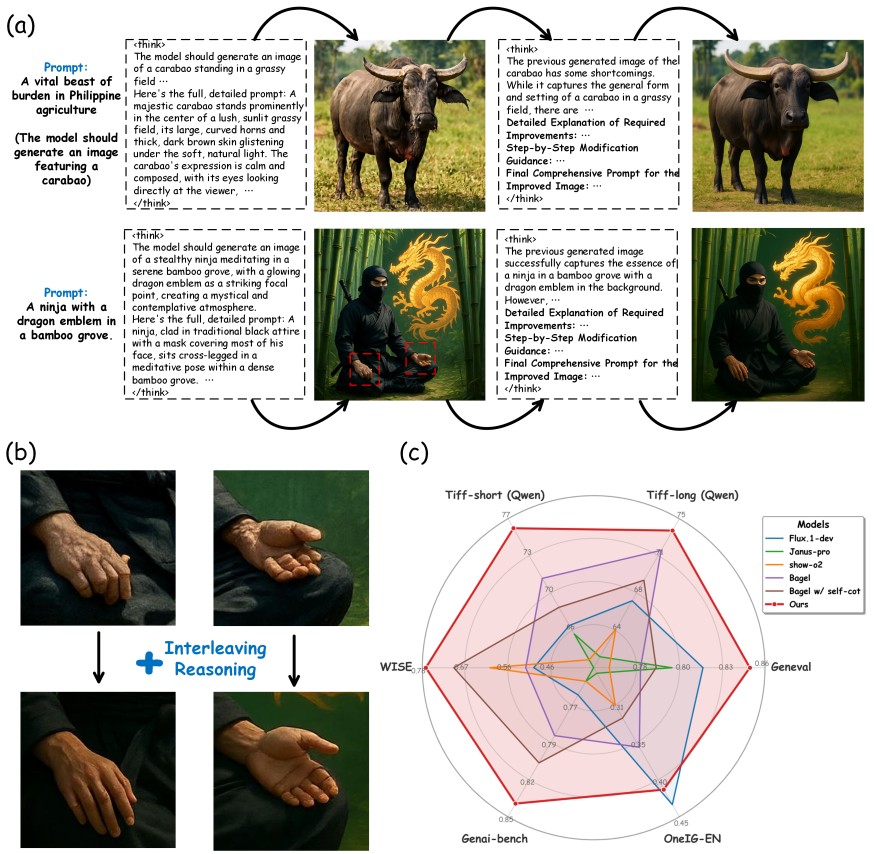

Figure 1: As shown in (a), we illustrate an example of Interleaving Reasoning Generation (IRG). Given a prompt, the model first produces a text-based reasoning process and then generates an image conditioned on that reasoning. Next, building upon the initial image, the model reflects on how to improve its quality and produces a refined image through this reflection process. IRG can substantially enhance image generation quality. For instance, in the top case of (a), IRG improves upon the previous generated image via multi-turn reasoning, enhancing rendering textures, shadow realism, and other visual properties. In the bottom case of (a), IRG significantly improves fine-grained details, such as the delicate structures of fingers—highlighted within the red box (as detailed in (b)). As shown in (c), compared to current SoTA models, our proposed IRG achieves clearly superior performance across multiple mainstream T2I benchmarks.

---

✉Corresponding authors.

ABSTRACT

Unified multimodal understanding and generation models recently have achieve significant improvement in image generation capability, yet a large gap remains in instruction following and detail preservation compared to systems that tightly couple comprehension with generation such as GPT-4o. Motivated by recent advances in interleaving reasoning, we explore whether such reasoning can further improve Text-to-Image (T2I) generation. We introduce *Interleaving Reasoning Generation (IRG)*, a framework that alternates between text-based thinking and image synthesis: the model first produces a text-based thinking to guide an initial image, then reflects on the result to refine fine-grained details, visual quality, and aesthetics while preserving semantics. To train IRG effectively, we propose *Interleaving Reasoning Generation Learning (IRGL)*, which targets two sub-goals: (1) strengthening the initial think-and-generate stage to establish core content and base quality, and (2) enabling high-quality textual reflection and faithful implementation of those refinements in a subsequent image. We curate *IRGL-300K*, a 300K-scale dataset organized into six decomposed learning modes that jointly cover learning text-based thinking, and full thinking–image trajectories. Starting from a unified foundation model that natively emits interleaved text–image outputs, our two-stage training first builds robust thinking and reflection, then efficiently tunes the IRG pipeline in the full thinking–image trajectory data. Extensive experiments show SoTA performance, yielding absolute gains of **5–10 points** on GenEval, WISE, TIIF, GenAI-Bench, and OneIG-EN, alongside substantial improvements in visual quality and fine-grained fidelity. As an early exploration, our results demonstrate that interleaving reasoning is a powerful paradigm for advancing T2I. The code, model weights and datasets will be released in: `https://github.com/Osilly/Interleaving-Reasoning-Generation`.

# 1 INTRODUCTION

Unified multimodal understanding and generation models consolidate image/text understanding and image synthesis capabilities within a single foundation model, and have recently emerged as a focal point of interest in the research community (Sun et al., 2024; Team, 2024; Tong et al., 2024; Wu et al., 2025a; Xie et al., 2024; Chen et al., 2025a; Xiao et al., 2025b;a; Liao et al., 2025; Xie et al., 2025b; Wu et al., 2025b; Deng et al., 2025). Representative efforts in this line of research, exemplified by GPT-4o (OpenAI, 2025c), seamlessly integrate comprehension and generation capabilities. This enables a pronounced performance gap relative to existing unified models, particularly in instruction-following for image generation and in the preservation of visual details (Deng et al., 2025).

Motivated by recent advances in text-based reasoning, notably test-time scaling techniques for (Multimodal) Large Language Models ((M)LLMs) (Jaech et al., 2024; Guo et al., 2025a; Huang et al., 2025; Chen et al., 2025b), a growing body of work has explored whether incorporating such text-based reasoning processes can yield improvements in the fidelity and overall quality of image generation (Fang et al., 2025a; Xiao et al., 2025b; Deng et al., 2025; Jiang et al., 2025). These works underscore this perspective and seek to exploit large-scale interleaved text–image corpora to learn subtle cross-modal interaction patterns, thus enabling seamless knowledge transfer between the understanding and generation stages of the model. However, in Text-to-Image (T2I) tasks, they employ only a single textual segment as auxiliary supervision in T2I generation, with the objective of producing outputs that more faithfully adhere to the original prompt. Recently, some works in (M)LLM field have focused on the *interleaving reasoning*, *i.e.*, multi-turn interactions and exhibits sophisticated reasoning dynamics (Huang, 2025), while this reasoning modality has empirical demonstrated superior accuracy in addressing complex problems (OpenAI, 2025b;a). This observation motivates the exploration:

*Whether interleaving reasoning can further enhance T2I generation quality?*

As shown in Fig 1, in generating a prompt-aligned image, the model is typically able to produce content that is broadly correct in terms of semantics, yet it remains challenging to attain superior visual quality and fine-grained fidelity (*e.g.*, in rendering textures, shadow realism, and delicate

structures such as fingers). The idea is intuitive: if high-quality image synthesis is considered a hard problem, adopting a multi-step reasoning strategy to tackle it is both reasonable and necessary. To solve this, a straightforward idea is to have the model first produce a text-based thinking process and generate an image based on that reasoning. Then, building on the initial image, the model reflects on how to improve its quality, and produces an improved image through reflection. We denote this process as ***Interleaving Reasoning Generation (IRG)***. Thus, we argue two points, 1) an additional text-based reasoning process can serve as auxiliary supervision for image generation, thereby alleviating the difficulty of direct generation, and 2) producing one image that simultaneously attains high visual quality and precise instruction following is non-trivial, whereas a multi-turn generation strategy can incrementally refine the output toward the desired goal. While these positions are aligned with prior reflection-based T2I generation approaches, the key distinction lies in their goals: prior methods (Tian et al., 2024; Zhuo et al., 2025; Wu et al., 2025b; Chern et al., 2025) generally employ reflection to rectify major semantic or structural errors in the generated content, with some adopting non-end-to-end frameworks, whereas our approach focuses on leveraging reflection to refine fine-grained details and improve overall visual quality in an end-to-end manner, with the main subject matter established during the initial generation. Specifically, through IRG, we not only enhance the semantic correctness of the generated content, but also focus on improving the quality, fine-grained details, and aesthetic aspects of the generated images.

Based on the insights, we firstly select a unified multimodal understanding and generation model as the base model, given its capability to produce interleaved text–image outputs. To facilitate interleaving reasoning-based generation with reflection, we propose the ***Interleaving Reasoning Generation Learning (IRGL)*** paradigm and formulate the objective as two sub-goals. The first is to strengthen the model's initial thinking and generation stage, which establishes the core content and base quality of the generated image. The second is to equip the model with the ability to produce detailed, high-quality text-based reflections, and to generate enhanced images that faithfully implement those reflections. In particular, we propose the ***IRGL-300K*** dataset, which refines the aforementioned two objectives into two complementary focuses: learning text-based thinking and mastering the image generation with thinking pipeline. It comprises *six decomposed learning modes* that jointly target comprehensive enhancement of model performance throughout the IRG process, while ensuring optimal exploitation of data resources. In the training pipeline, the model is first trained to generate accurate initial thinking given a prompt, as well as to produce text-based reflections based on the initial reasoning step that improve quality. Furthermore, we incorporate full thinking–image trajectories to mitigate potential degradation in the model's core generative ability in this training stage. Then, leveraging the acquired thinking generation capability, we utilize the full thinking–image trajectory data to efficiently train the entire IRG pipeline.

Extensive experiments demonstrate that our proposed IRG achieves State-of-The-Art (SoTA) benchmark performance, delivering absolute improvements of 5–10 points across multiple benchmarks, including GenEval, WISE, TIIF, GenAI-Bench, and OneIG-EN. In addition, IRG significantly enhances visual quality and fine-grained fidelity. As an early effort to introduce Interleaving Reasoning into the T2I domain, we hope our work can inspire future research in this direction.

## 2 METHOD

### 2.1 PRELIMINARIES AND NOTATIONS

Unified multimodal understanding and generation models based on an integrated transformer architecture, *e.g.*, BAGEL (Deng et al., 2025), jointly perform image understanding and generation within a single architecture, facilitated by large-scale training on interleaved text–image data. Such models can naturally handle both interleaved inputs and interleaved outputs. Let $T_{in}$ and $I_{in}$ represent the input text and image, respectively, while $T_{out}^{(1)}$ and $I_{out}^{(1)}$ denote the initial output text and image, respectively. The standard image understanding and T2I generation with self-CoT reasoning process can be formulated as:

$$\text{Image understanding:} \quad T_{in} + I_{in} \rightarrow T_{out}^{(1)},$$
$$\text{T2I generation with self-CoT:} \quad T_{in} \rightarrow T_{out}^{(1)} \rightarrow I_{out}^{(1)}. \tag{1}$$

Previous works that adopt self-CoT (Deng et al., 2025; Fang et al., 2025a; Xiao et al., 2025b; Jiang et al., 2025) typically focus solely on text-based reasoning to improve image generation, while

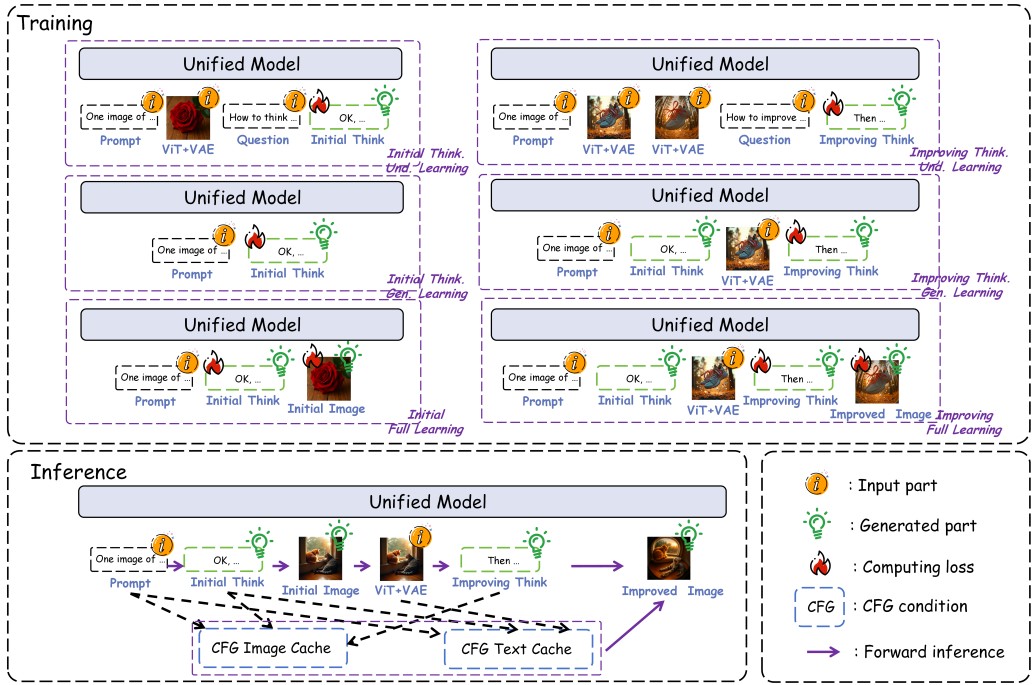

Figure 2: Overview of our proposed IRG training and inference pipeline. IRG learns the text-based thinking process and the complete high-quality image generation pipeline under six decomposed learning modes. During inference, we introduce a dedicated CFG condition design (Ho & Salimans, 2022) for IRG's improved image generation steps. The "Input part" denotes the inputs to the IRG module, comprising the original prompt, ViT and VAE features of the image, and our constructed queries. "The Generated part" refers to the outputs produced by IRG, including the intermediate text-based reasoning process and the synthesized image. "Computing loss" indicates which components participate in the loss computation during training.

overlooking the potential of leveraging the initially generated image to further enhance visual quality and perform multi-step information fusion for better results. Thus, to conduct additional T2I reasoning step conditioned on the previously generated text and image, we first encode the generated image into Vision Transformer (ViT) features and Variational Autoencoder (VAE) features, which are defined as $I_f$. Then, based on the full information cache, the multi-turn generation can be conducted and the pipeline denoted as follows:

$$T_{in} \rightarrow T_{out}^{(1)} \rightarrow I_{out}^{(1)} \xrightarrow{\text{enc}} I_f^{(1)} \rightarrow T_{out}^{(2)} \rightarrow I_{out}^{(2)} \xrightarrow{\text{enc}} I_f^{(2)} \rightarrow \cdots \rightarrow T_{out}^{(n)} \rightarrow I_{out}^{(n)}, \quad (2)$$

where $n$ means the total turn number and $\xrightarrow{\text{enc}}$ represents the image feature encoding process, and note that for intermediate images generated at the $k$-th turn (e.g., $I_{out}^{(k)}$), only their encoded representations $I_f^{(k)}$ are propagated to the subsequent computation stages. During the generation of the final image $I_{out}^{(n)}$, the model exchanges and exploits multiple segments of interleaved text–image representations, a process we term **Interleaving Reasoning Generation (IRG)**. By formulating the synthesis of the final image as the ultimate goal, the model employs multi-stage progressive generation and cross-modal information fusion to maximize output quality. We note that an inherent strength of unified models lies in their capacity to process and generate interleaved inputs and outputs, inherently supporting multi-turn generation with self-CoT reasoning. This property enables us to investigate how interleaving reasoning within the generation pipeline can further extend the achievable limits of generative capability.

## 2.2 INTERLEAVING REASONING GENERATION

### 2.2.1 OVERVIEW

As mentioned above, IRG can be defined as comprising two components: (1) an initial text-based reasoning process followed by image generation based on that reasoning, and (2) repetition of the

first component to produce an improved image. In this work, we focus solely on a single refinement iteration, *i.e.*, limiting the second component to one turn (set $n$ in Eq. 2 to 2), in order to validate our hypothesis on whether Interleaving Reasoning can effectively enhance text-to-image generation quality.

In the following sections, we detail the ***Interleaving Reasoning Generation Learning (IRGL)*** framework and explain how different forms of interleaving reasoning data can be effectively utilized to perform hierarchical learning with distinct emphases, as elaborated in Sec. 2.2.2. Furthermore, in Sec. 2.2.3, we introduce the data construction pipeline of ***IRGL-300k*** dataset, while in Sec. 2.2.4, we simply describe the inference strategy of IRG, such as the Classifier-Free Guidance (CFG) (Ho & Salimans, 2022) condition designs.

### 2.2.2 INTERLEAVING REASONING GENERATION LEARNING

When we set $n$ to 2, Eq. 2 will be reformulate as:

$$T_{in} \rightarrow T_{out}^{(1)} \rightarrow I_{out}^{(1)} \xrightarrow{\text{enc}} I_f^{(1)} \rightarrow T_{out}^{(2)} \rightarrow I_{out}^{(2)}. \tag{3}$$

It can be observed that when aiming to enhance the quality of the final image $I_{out}^{(2)}$ in IRG pipeline, we decompose the process into four progressive intermediate steps: (1) ensuring the correctness of the initial thinking process $T_{out}^{(1)}$, (2) improving the quality of the initial generated image $I_{out}^{(1)}$, (3) generating an accurate improving-thinking step $T_{out}^{(2)}$ based on the first image to guide the production of a better image, and (4) synthesizing the final high-quality image $I_{out}^{(2)}$ by integrating all preceding decision steps. Evidently, as shown in Fig 2, various decomposed learning modes can be designed to improve the model's intermediate reasoning capacity. We begin with the enhancement of the initial reasoning step, *i.e.*, above steps (1) and (2), which can be instantiated as the following tasks:

- **Initial Thinking Understanding Learning:** In this task, we aim for the model to learn how to generate the correct initial thinking process given original prompt $T_{in}$ and prior image features $I_f^{(1)}$. The design insight behind this task is as follows: when the model is provided with both a prompt and an image consistent with that prompt, we construct an auxiliary question $T_q$ in which the model learns, through image-understanding supervision, how to produce a reasoning process aligned with the prompt and to recognize what kind of image such a reasoning process would yield.

$$T_{in} + I_f^{(1)} + T_q \rightarrow T_{out}^{(1)}. \tag{4}$$

- **Initial Thinking Generation Learning:** This task directly imitates the reasoning process for generating initial thinking given the original prompt, and can be considered a more challenging task compared to Initial Thinking Understanding Learning.

$$T_{in} \rightarrow T_{out}^{(1)}. \tag{5}$$

- **Initial Full Learning:** In this full initial reasoning learning setting, the model learns from both text-based reasoning sequences and high-quality image data to enhance the quality of its initial image generation, thereby providing a stronger foundation for producing improved images in subsequent reasoning stages.

$$T_{in} \rightarrow T_{out}^{(1)} \rightarrow I_{out}^{(1)}. \tag{6}$$

Furthermore, we also design three tasks to learning how to generate the improving thinking and improved image based on the initial reasoning step (the above steps (3) and (4)):

- **Improving Thinking Understanding Learning:** This task is closely related to Initial Thinking Understanding Learning, but focuses on enabling the model to, given a prompt, learn to generate the improving thinking process for enhancing an initial image to an improved image. This is achieved by understanding the prompt $T_{in}$, comparing the differences between the features of the initial $I_f^{(1)}$ and improved images $I_f^{(2)}$ and by answering carefully designed questions $T_q$.

$$T_{in} + I_f^{(1)} + I_f^{(2)} + T_q \rightarrow T_{out}^{(2)}. \tag{7}$$

- **Improving Thinking Generation Learning:** This task builds on the initial reasoning stage, focusing on learning how to generate improving thinking.

$$T_{in} + T_{out}^{(1)} + I_f^{(1)} \rightarrow T_{out}^{(2)}. \tag{8}$$

- **Improving Full Learning:** This task represents a complete IRG process, but we constrain the model, under the condition that the initial reasoning is already completed, to learn only the improving reasoning and the high-quality improved image components. As the most crucial stage of IRG, the model must learn to identify the differences in visual quality and fine-grained fidelity between the two images, and to leverage this understanding to generate the optimal image during the improving reasoning step.

$$T_{in} + T_{out}^{(1)} + I_f^{(1)} \rightarrow T_{out}^{(2)} \rightarrow I_{out}^{(2)}. \tag{9}$$

The aforementioned decomposed learning modes can be clearly divided into two training objectives: (1) learning the text-based thinking process (Initial Thinking Understanding Learning, Initial Thinking Generation Learning, Improving Thinking Understanding Learning, and Improving Thinking Generation Learning), and (2) learning the complete high-quality image generation pipeline under the auxiliary supervision of the reasoning process (Initial Full Learning and Improving Full Learning). This design likewise addresses the limited availability of high-quality, full IRG thinking-image trajectories data, while learning from text-based reasoning serves as a partial remedy to this problem.

Furthermore, we employ a two-stage training pipeline. In stage 1, the model is optimized on all six tasks to generate accurate initial reasoning from a given prompt and to produce text-based reflections derived from the initial reasoning step to enhance output quality. The main goal of this stage is to strengthen the text-based reasoning capability, while incorporating full thinking–image trajectories to avoid degrading the core generative performance. Empirically, we find that this reasoning-focused training converges relatively rapidly. In the second stage, leveraging the thinking generation ability learned in Stage 1, we employ full thinking–image trajectory data (*i.e.*, data in Initial Full Learning and Improving Full Learning) to efficiently optimize the entire IRG pipeline. In this training stage involving image generation, convergence demands more iterations, since the model must spend additional training time learning the fine-grained fidelity transformations from the initial image to the improved image.

**Discussion:** Unfortunately, constructing complete interleaving IRG data is non-trivial, even when considering only the two-turn case, due to the following two challenges. First, obtaining final high-quality images is inherently difficult, as the quality of existing open-source T2I datasets remains suboptimal. This limitation has motivated many recent works to distill images generated by powerful models such as GPT-4o (Chen et al., 2025a; Wu et al., 2025b). Second, although a subset of high-quality data from GPT-4o is available, IRG requires paired samples linking an initial image to its improved counterpart. Designing the transformation process from the initial image to the improved image is itself a challenge, which means that such pairs must be constructed from scratch and cannot be directly derived from GPT-4o-distilled T2I data. These two issues make it difficult to obtain complete IRG datasets at scale. To mitigate the scarcity of fully optimized training data, we design multiple intermediate training objectives. This is because we only learn the text-based thinking process during intermediate objective training, in order to avoid low-quality image data pollution. We expect that, when conditions permit, access to a large quantity of complete IRG data would lead to even better performance.

### 2.2.3 INTERLEAVING REASONING DATA CONSTRUCTION

This section introduces the IRGL-300k dataset construction pipeline for the aforementioned six decomposed learning modes. We utilize one MLLM and the strong image generation model to construct the dataset for training.

**Data for learning the initial reasoning step.** For the Initial Thinking Understanding Learning task, we construct training data from open-source T2I datasets containing prompt–image pairs. First, we design an initial thinking template, and then instruct a large language model (*e.g.*, Qwen2.5-VL (Bai et al., 2025b)) to generate a reasoning process that is consistent with both the prompt and the corresponding image from the original T2I data. Finally, we organize the data according to Eq. 4: the prompt corresponds to $T_{\text{in}}$, the image is encoded to obtain features $I_f^{(1)}$, a manually designed

Table 1: **Evaluation of text-to-image generation ability on GenEval benchmark.** 'Gen. Only' stands for an image generation model, and 'Unified' denotes a model that has both understanding and generation capabilities. † refer to the methods using MLLM rewriter. * means we report the reproducing results using the official Github repository and checkpoint. The best Overall results are **bolded**.

| Type | Model | Single Obj. | Two Obj. | Counting | Colors | Position | Color Attri. | Overall↑ |
|------|-------|-------------|----------|----------|--------|----------|--------------|----------|
| Gen. Only | PixArt-$\alpha$ (Chen et al., 2024) | 0.98 | 0.50 | 0.44 | 0.80 | 0.08 | 0.07 | 0.48 |
| | SDv2.1 (Rombach et al., 2022) | 0.98 | 0.51 | 0.44 | 0.85 | 0.07 | 0.17 | 0.50 |
| | DALL-E 2 (Ramesh et al., 2022) | 0.94 | 0.66 | 0.49 | 0.77 | 0.10 | 0.19 | 0.52 |
| | Emu3-Gen (Wang et al., 2024b) | 0.98 | 0.71 | 0.34 | 0.81 | 0.17 | 0.21 | 0.54 |
| | SDXL (Podell et al., 2024) | 0.98 | 0.74 | 0.39 | 0.85 | 0.15 | 0.23 | 0.55 |
| | DALL-E 3 (Betker et al., 2023) | 0.96 | 0.87 | 0.47 | 0.83 | 0.43 | 0.45 | 0.67 |
| | SD3-Medium (Esser et al., 2024) | 0.99 | 0.94 | 0.72 | 0.89 | 0.33 | 0.60 | 0.74 |
| | FLUX.1-dev$^\dagger$ (Labs, 2024) | 0.98 | 0.93 | 0.75 | 0.93 | 0.68 | 0.65 | 0.82 |
| Unified | Chameleon (Team, 2024) | - | - | - | - | - | - | 0.39 |
| | LWM (Liu et al., 2024) | 0.93 | 0.41 | 0.46 | 0.79 | 0.09 | 0.15 | 0.47 |
| | SEED-X (Ge et al., 2024) | 0.97 | 0.58 | 0.26 | 0.80 | 0.19 | 0.14 | 0.49 |
| | TokenFlow-XL (Qu et al., 2024) | 0.95 | 0.60 | 0.41 | 0.81 | 0.16 | 0.24 | 0.55 |
| | ILLUME (Wang et al., 2024a) | 0.99 | 0.86 | 0.45 | 0.71 | 0.39 | 0.28 | 0.61 |
| | Janus (Wu et al., 2025a) | 0.97 | 0.68 | 0.30 | 0.84 | 0.46 | 0.42 | 0.61 |
| | Transfusion (Zhou et al., 2024) | - | - | - | - | - | - | 0.63 |
| | Emu3-Gen$^\dagger$ (Wang et al., 2024b) | 0.99 | 0.81 | 0.42 | 0.80 | 0.49 | 0.45 | 0.66 |
| | Show-o (Xie et al., 2024) | 0.98 | 0.80 | 0.66 | 0.84 | 0.31 | 0.50 | 0.68 |
| | Janus-Pro-7B (Chen et al., 2025c) | 0.99 | 0.89 | 0.59 | 0.90 | 0.79 | 0.66 | 0.80 |
| | MetaQuery-XL$^\dagger$ (Pan et al., 2025) | - | - | - | - | - | - | 0.80 |
| | BAGEL* (Deng et al., 2025) | 0.99 | 0.95 | 0.76 | 0.87 | 0.50 | 0.60 | 0.78 |
| | Show-o2 (Xie et al., 2025b) | 1.00 | 0.87 | 0.58 | 0.92 | 0.52 | 0.62 | 0.76 |
| | BAGEL $w/$ self-CoT* (Deng et al., 2025) | 0.99 | 0.92 | 0.75 | 0.89 | 0.54 | 0.63 | 0.79 |
| | IRG (Ours) | 0.98 | 0.94 | 0.83 | 0.86 | 0.74 | 0.73 | **0.85** |
| | GPT-4o (OpenAI, 2025c) | 0.99 | 0.92 | 0.85 | 0.92 | 0.75 | 0.61 | 0.84 |

understanding question (*e.g.*, "You have been given one prompt and one image ...") corresponds to $T_q$, and the MLLM-generated initial thinking corresponds to $T_{\text{out}}^{(1)}$.

For the Initial Thinking Generation Learning, the initial thinking acquisition pipeline is similar to the Initial Thinking Understanding Learning task. It uses the prompt and MLLM-generated initial thinking to obtain the train date (as Eq. 5).

For the Initial Full Learning data, to ensure learning from high-quality image information, we input the original prompt into a high-quality image generation model (GPT-4o (OpenAI, 2025d)) to produce a high-quality image, which serves as $I_{\text{out}}^{(1)}$ in Eq. 6. The initial thinking is obtained in a similar manner to that described above, by providing the prompt and the GPT-4o-distilled high-quality image to an MLLM.

**Data for learning the improving reasoning step.** For generating data for the improving reasoning step, we encounter a key challenge: given access to a high-quality image to serve as the improved image, we must determine the source of the initial reasoning step data. We choose to use data generated by the base model (*i.e.*, BAGEL (Deng et al., 2025)) conditioned on the same prompt as the initial reasoning step data. This design decision is motivated by two considerations: (1) it provides a simple and efficient way to obtain multi-turn IRG data at scale, and (2) it allows us to improve the model's performance without compromising the original capabilities of the base model.

For the Improving Thinking Understanding Learning task, we first input the prompt into the base model to generate the initial thinking and the corresponding initial image. We then design an instructional prompt to guide the MLLM in generating improved thinking based on the base model's generated image and the image from the T2I dataset. The model is instructed to produce the improving thinking according to a predefined template. We adopt a stage-level template (Xu et al., 2024), which requires the model to first generate a part of the analysis of previous generated image issues. Then, generate the stages in the format "Detailed Explanation of Required Improvements: ...", "Step-by-Step Modification Guidance: ...", and "Final Comprehensive Prompt for the Improved Image: ...". Finally, we organize the original prompt $T_{in}$ and initial image generated by base model $I_f^{(1)}$, the image in the T2I dataset $I_f^{(2)}$, a manually designed understanding question (*e.g.*, "You have been given one prompt and two images ...") $T_q$, and the improving thinking $T_{out}^{(2)}$ by Eq. 7.

Table 2: **Comparison of world knowledge reasoning on WISE.** WISE examines the complex semantic understanding and world knowledge for T2I generation. 'Gen. Only' stands for an image generation model, and 'Unified' denotes a model that has both understanding and generation capabilities. * means we report the reproducing results using the official Github repository and checkpoint. The best results are **bolded**.

| Type | Model | Cultural | Time | Space | Biology | Physics | Chemistry | Overall↑ |
|------|-------|----------|------|-------|---------|---------|-----------|----------|
| *Gen. Only* | SDv1.5 (Rombach et al., 2022) | 0.34 | 0.35 | 0.32 | 0.28 | 0.29 | 0.21 | 0.32 |
| | SDXL (Podell et al., 2024) | 0.43 | 0.48 | 0.47 | 0.44 | 0.45 | 0.27 | 0.43 |
| | SD3.5-large (Esser et al., 2024) | 0.44 | 0.50 | 0.58 | 0.44 | 0.52 | 0.31 | 0.46 |
| | PixArt-Alpha (Chen et al., 2024) | 0.45 | 0.50 | 0.48 | 0.49 | 0.56 | 0.34 | 0.47 |
| | playground-v2.5 (Li et al., 2024b) | 0.49 | 0.58 | 0.55 | 0.43 | 0.48 | 0.33 | 0.49 |
| | FLUX.1-dev (Labs, 2024) | 0.48 | 0.58 | 0.62 | 0.42 | 0.51 | 0.35 | 0.50 |
| *Unified* | Janus (Wu et al., 2025a) | 0.16 | 0.26 | 0.35 | 0.28 | 0.30 | 0.14 | 0.23 |
| | VILA-U (Wu et al., 2024d) | 0.26 | 0.33 | 0.37 | 0.35 | 0.39 | 0.23 | 0.31 |
| | Show-o-512 (Xie et al., 2024) | 0.28 | 0.40 | 0.48 | 0.30 | 0.46 | 0.30 | 0.35 |
| | Janus-Pro-7B (Chen et al., 2025c) | 0.30 | 0.37 | 0.49 | 0.36 | 0.42 | 0.26 | 0.35 |
| | Emu3 (Wang et al., 2024b) | 0.34 | 0.45 | 0.48 | 0.41 | 0.45 | 0.27 | 0.39 |
| | MetaQuery-XL (Pan et al., 2025) | 0.56 | 0.55 | 0.62 | 0.49 | 0.63 | 0.41 | 0.55 |
| | BAGEL (Deng et al., 2025) | 0.44 | 0.55 | 0.68 | 0.44 | 0.60 | 0.39 | 0.52 |
| | Show-o2* (Xie et al., 2025b) | 0.64 | 0.58 | 0.61 | 0.58 | 0.63 | 0.49 | 0.61 |
| | BAGEL *w/* self-CoT (Deng et al., 2025) | 0.76 | 0.69 | 0.75 | 0.65 | 0.75 | 0.58 | 0.70 |
| | IRG (Ours) | **0.78** | **0.72** | **0.76** | **0.81** | **0.82** | **0.78** | **0.77** |
| | GPT-4o (OpenAI, 2025c) | 0.81 | 0.71 | 0.89 | 0.83 | 0.79 | 0.74 | 0.80 |

For the Improving Thinking Generation Learning task, the data is same as Improving Thinking Understanding Learning, while it use the initial thinking generated by base model as $T_{out}^{(1)}$ in Eq. 8

In the Improving Full Learning setting, GPT-4o is used to produce the improved image in the IRG trajectory. Given the prompt and the initial image from the base model, GPT-4o generates a higher-quality, prompt-consistent image, which we adopt as the improved image $I_{out}^{(2)}$ in Eq. 9. An MLLM is then employed to produce stage-level improving thinking detailing the transformation from the initial image to the improved image.

### 2.2.4 INFERENCE STRATEGY

As shown in Fig 2, the model produces a "text–image–text–image" trajectory. This poses a challenge: in conventional diffusion-based generation models, the CFG-conditioning design is typically straightforward, such as directly comparing the presence versus absence of the prompt. In contrast, for our proposed IRG under even only two-turn reasoning pipeline, particularly before generating the improved image—there are four possible condition sources to compare (*i.e.*, the prompt, the initial reasoning, the initial image, and the improving reasoning). Therefore, a customized CFG-conditioning strategy is required. Based on this, we adopt the framework incorporates two complementary CFG condition schemes: (1) conditioning with versus without the image information from the initial generation, and (2) conditioning with versus without the reflection text. In practice, we set the the guidance scale hyper-parameters in CFG Image condition (versus without the image information) and CFG text condition (versus without the text information) to 2.0. This strategy helps maintain high visual quality and fidelity in images produced during the improving reasoning steps, with notable benefits for generation stability (see Fig 5).

## 3 EXPERIMENTS

Please refer to Appendix B to obtain the detailed data and training settings. We provide the main results of IRG in Sec. 3.1 and Appendix C, and then we introduce the detailed ablation study, the result analysis and the visualization in Sec. 3.2 and Appendix D.

### 3.1 MAIN RESULTS

Across mainstream benchmarks (*e.g.*, GenEval (Ghosh et al., 2023) and WISE (Niu et al., 2025)), IRG consistently surpasses strong unified baselines and self-CoT variants, evidencing benefits of interleaving reasoning with generation beyond any single metric suite. On GenEval (Tab. 1), IRG achieves the best Overall 0.85 (+0.07/+0.06 over BAGEL w/ self-CoT), with strong subskills in

Table 3: **Ablation study of IRG.** The base method is BAGEL w/ self-CoT. "High-quality Images Training" means that we directly use the image data of Initial Full Learning and Improving Full Learning (see Sec. 2.2.2) to train the base model. "Interleaving Reasoning Generation" indicates that using the full IRG thinking-image trajectories during training. "Decomposed Learning Modes" means that we train the model in decomposed learning modes with two-stage training.

| Model | WISE | TIIF | GenAI-Bench |
|---|---|---|---|
| BAGEL $w/$ self-CoT (Deng et al., 2025) | 0.70 | 68.06/68.78 | 0.81 |
| + High-quality Images Training | 0.73 | 70.69/69.85 | 0.80 |
| + Interleaving Reasoning Generation | 0.76 | 73.90/71.37 | 0.83 |
| + Decomposed Learning Modes (Ours) | 0.77 | 76.00/73.77 | 0.84 |

Table 4: **Analysis of single-turn and multi-turn IRG pipeline.** "IRG reasoning step 1" means that the initial image generated by IRG as the evaluation images.

| Model | WISE | | | | | | TIIF | GenAI-Bench |
|---|---|---|---|---|---|---|---|---|
| | Score | Rank score | | | | | Score | Score |
| | | Qwen | GPT-4o | UnifiedReward | Avg. | Human | | |
| IRG reasoning step 1 | 0.79 | 29% ↓ | 38% ↓ | 43% ↓ | 36.7% ↓ | 17% ↓ | 75.84/73.90 | 0.84 |
| IRG (Ours) | 0.77 | 71% ↑ | 62% ↑ | 57% ↑ | 63.3% ↑ | 74% ↑ | 76.00/73.77 | 0.84 |

Counting 0.83 and Position 0.74. On WISE (Tab. 2), it sets a unified SOTA of 0.77 (+0.07 over BAGEL w/ self-CoT), with consistent gains across domains (*e.g.*, Biology 0.81, Physics 0.82). For more benchmark results, such as TIIF (Wei et al., 2025) (Tab. 5), GenAI-Bench (Li et al., 2024a) (Tab. 6), and OneIG-EN (Chang et al., 2025) (Tab. 7), with more detailed experiment analysis, are presented in Appendix C.

## 3.2 EXPERIMENTAL RESULTS ANALYSIS

**Ablation study.** Tab. 3 isolates the contributions of data and training objectives. Adding high-quality image training on top of a self-CoT baseline brings moderate improvements (WISE 0.70→0.73, and TIIF 68.06/68.78→70.69/69.85), while when introduce the IRG pipeline (*i.e.*, using the data of Initial Full Learning and Improving Full Learning) achieve the significant improvement (0.03 benchmark score improvement in both WISE and GenAI-Bench). Furthermore, adopting IRG with six decomposed learning modes yields the largest jump (WISE 0.77, TIIF 76.00/73.77, GenAI-Bench 0.84). This supports our hypothesis that text-only thinking supervision (Sec. 2.2.2) is a data-efficient proxy for scarce full IRG thinking-image trajectories, and that mixing text-based thinking learning, and full learning modes provides complementary signals.

**Analysis of single-turn and multi-turn IRG pipeline.** As shown in Tab. 4, two-turn IRG and the initial generated images achieve similar benchmark scores, and on some benchmarks, initial generated image even attains slightly higher scores. However, this does not imply that multi-turn IRG offers no benefits. Our proposed IRG leverages image-conditioned reflection to enhance visual quality and fine-grained fidelity.

We evaluate the initial generated image and improved image generated by IRG on the WISE benchmark by prompting a MLLM to directly compare the two images in terms of generation quality, fine-grained details, aesthetics, and other visual aspects. To eliminate positional bias, we randomly shuffle the order of the two images presented to the MLLM and repeat the evaluation three times, reporting the averaged results. Multiple MLLMs are employed as evaluators, including Qwen2.5-VL-72B (Bai et al., 2025b), GPT-4o (OpenAI, 2025c), and UnifiedReward (Wang et al., 2025b) (using its default pairwise comparison protocol, which incorporates the original task text prompt). This multiple MLLMs as judge method mitigates evaluator-specific biases and indicates better generalization of perceived quality.

The ranking study in Tab. 4 shows that the full IRG pipeline improves agreement with multiple automatic raters compared to the first-step-only variant (average rank score: 63.3% vs. 36.7%), suggesting that two-turn IRG produces images whose improvements are consistently recognized by heterogeneous MLLM evaluators.

Furthermore, we recruited three professional human annotators to evaluate the image quality generated by IRG at reasoning steps 1 and 2 on the WISE benchmark. The annotators were instructed to compare the quality of two images based on the original prompt, with the order of images from the two IRG

rounds randomly shuffled. They were allowed to select from three options: "A is better", "B is better", or "Equally good". As shown in Tab. 4, the human evaluation results demonstrate strong consistency with the MLLM judge results. Notably, human judges exhibit an even stronger preference for images generated at IRG reasoning step 2, which validates that IRG can significantly enhance image generation quality through multi-turn reasoning.

## 4 CONCLUSION

In this paper, we propose the Interleaving Reasoning Generation (IRG) framework, which generates high-quality images through a "text–image–text–image" process. Specifically, given a prompt, the model first produces a text-based reasoning sequence and then generates an image conditioned on that reasoning. Next, building upon the initial image, the model reflects on how to improve its quality and produces a refined image through this reflection process. Extensive experiments on mainstream benchmarks demonstrate significant improvements in generation performance.

## ACKNOWLEDGEMENTS

This work is supported by the National Natural Science Foundation of China (NO. 62572193), the Open Research Fund of the Key Laboratory of Advanced Theory and Application in Statistics and Data Science, Ministry of Education, and the Fundamental Research Funds for the Central Universities, JC STEM Lab of AI for Science and Engineering, funded by The Hong Kong Jockey Club Charities Trust, the MTR Research Funding (MRF) Scheme (CHU-24003), the Research Grants Council of Hong Kong (Project No. CUHK14213224).

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

# A   RELATED WORK

## A.1   UNIFIED MULTIMODAL UNDERSTANDING AND GENERATION MODELS

Unified Multimodal Understanding and Generation Models has attracted much attention of the research community. The mainstream research works can be divide into three categories: (1) Autoregressive (Wu et al., 2025a; Chen et al., 2025c; Lu et al., 2024; Qu et al., 2024; Team, 2024; Wang et al., 2024b). These methods adopt the next token prediction paradigm to generate the text and image token in one unified model. (2) Additional Diffusion (Dong et al., 2024; Wu et al., 2024b; Pan et al., 2025; Tong et al., 2024; Gao et al., 2025). They usually combine a pre-trained LLM backbone with an external diffusion module. The LLM is then used to obtain the semantic conditions that enable the diffusion module to generate an image. And (3) Unified Integrated Transformer (Deng et al., 2025; Liang et al., 2024; Ma et al., 2024; Shi et al., 2024; Zhou et al., 2024). In this category, works typically integrate the LLM and diffusion models in one transformer model. Our proposed IRG based on the unified integrated transformer model BAGEL (Deng et al., 2025) due to it was pre-trained on the large -scale interleaved text–image data. In principle, our proposed IRG framework can be effectively applied to all of the aforementioned types of unified models, as they naturally handle both interleaved inputs and interleaved outputs.

## A.2   REASONING MODELS

Text-based reasoning models have achieved significant progress in solving a wide range of real-world tasks (Jaech et al., 2024; Guo et al., 2025a; Huang et al., 2025; Chen et al., 2025b;d; Zeng et al., 2025a; Gu et al., 2025; Li et al., 2025; You et al., 2025; Huang et al., 2026b; Zeng et al., 2026; Jia et al., 2025; Feng et al., 2025; Zhan et al., 2025; Wang et al., 2026; Tang et al., 2026; Yin et al., 2025; **?**; Zeng et al., 2025b; Qi et al., 2025; Zhao et al., 2025; Fang et al., 2025b; Wang et al., 2025a; Huang et al., 2026a). Recently, several works have begun to adopt interleaving reasoning to address more complex problems (Huang, 2025; OpenAI, 2025b;a), *i.e.*, incorporating non-text modalities into multi-turn reasoning processes. In the text-to-image (T2I) domain, the latest studies explore whether introducing a text-based reasoning step can enhance image generation performance (Fang et al., 2025a; Xiao et al., 2025b; Deng et al., 2025; Jiang et al., 2025). Our IRG framework is inspired by these advances and seeks to integrate interleaving reasoning into the T2I generation process. While our approach shares conceptual similarities with reflection-based T2I methods (Tian et al., 2024; Zhuo et al., 2025; Wu et al., 2025b; Chern et al., 2025), the key distinction lies in IRG's dual focus: not only enhancing the semantic correctness of generated content but also improving visual quality, fine-grained fidelity, and aesthetic appeal. Furthermore, compared to Self-Correcting with LLM (Wu et al., 2024c) and CoT-based Image Generation (Guo et al., 2025b) methods, our proposed IRG offers three key advantages: (1) it integrates the reflection process into an end-to-end framework instead of relying on external reward models or LLMs for judgment; (2) it introduces a novel "text-image-text-image" interleaved reasoning framework that leverages text-based reasoning to guide image generation, rather than solely focusing on image-based reflection; (3) through broad paradigm learning, it achieves significant improvements across multiple diverse benchmarks in both semantic correctness and visual quality.

# B   EXPERIMENT SETTINGS

**IRGL-300k dataset.**   For the Initial Thinking Understanding Learning, Initial Thinking Generation Learning, Improving Thinking Understanding Learning, and Improving Thinking Generation Learning tasks, we use the open-source T2I dataset OSP1024-286k (Lin et al., 2025). For each of the four tasks, we sample 50K instances from the dataset for data construction.

We employ the GPT-4o-distilled T2I dataset BLIP3o-60k (Chen et al., 2025a) in Initial Full Learning. For Improving Full Learning, we construct the dataset by distilling GPT-4o with a curated set of prompts. Specifically, this prompt set is sourced from the training set of T2I-compbench (following (Jiang et al., 2025)). In addition, for each prompt, we use Qwen3 (Yang et al., 2025) to generate 1–2 complex prompt variants. Furthermore, following (Chen et al., 2025a), we construct prompts from common entities and then use Qwen3 to rewrite each into 2–3 complex variants (we do not use the original entity prompts directly to obtain the GPT-4o-distilled images). In total, this procedure produces approximately 30K samples.

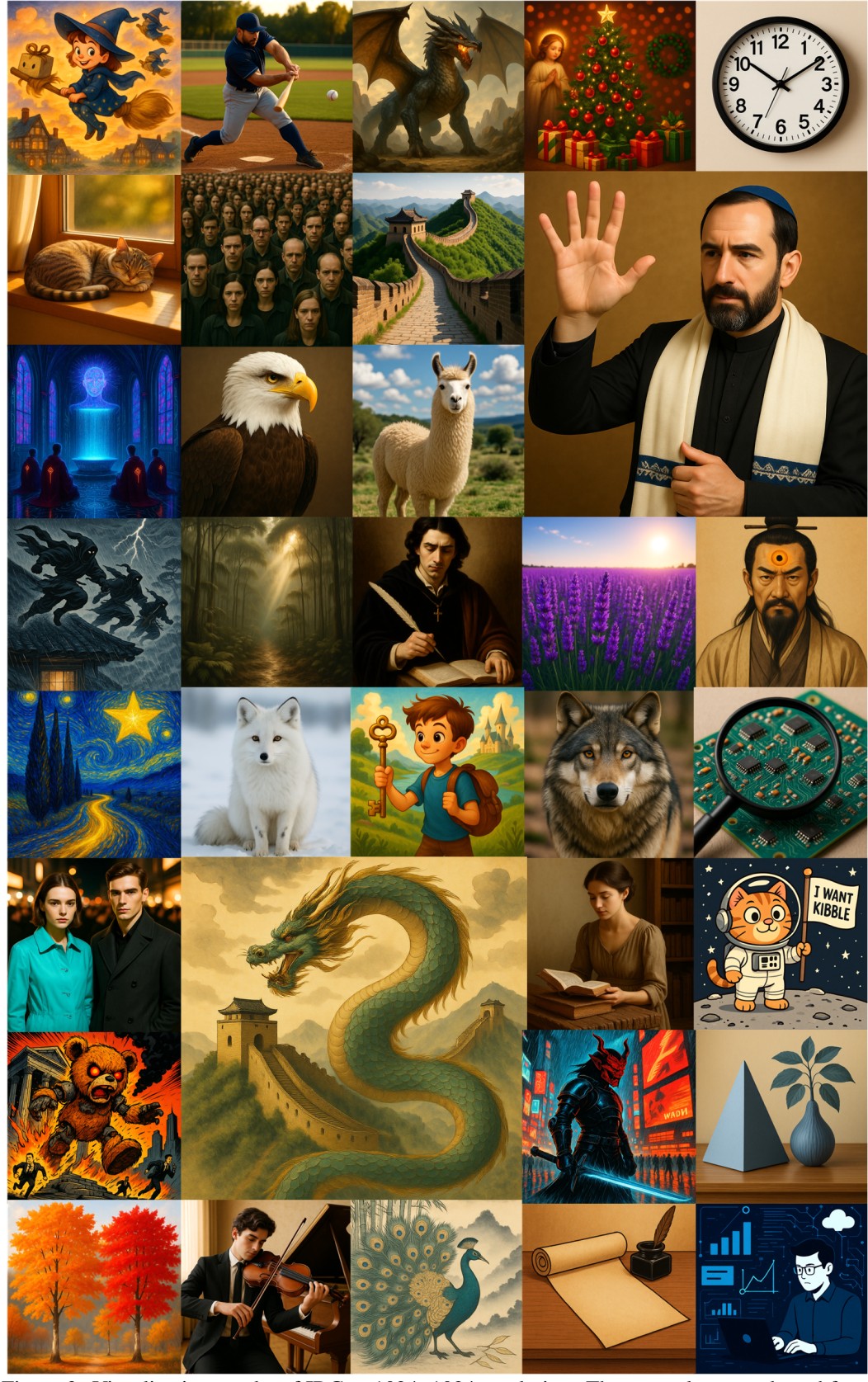

Figure 3: Visualization results of IRG at 1024×1024 resolution. The examples are selected from WISE (Niu et al., 2025), TIIF (Wei et al., 2025), and GenAI-Bench (Li et al., 2024a).

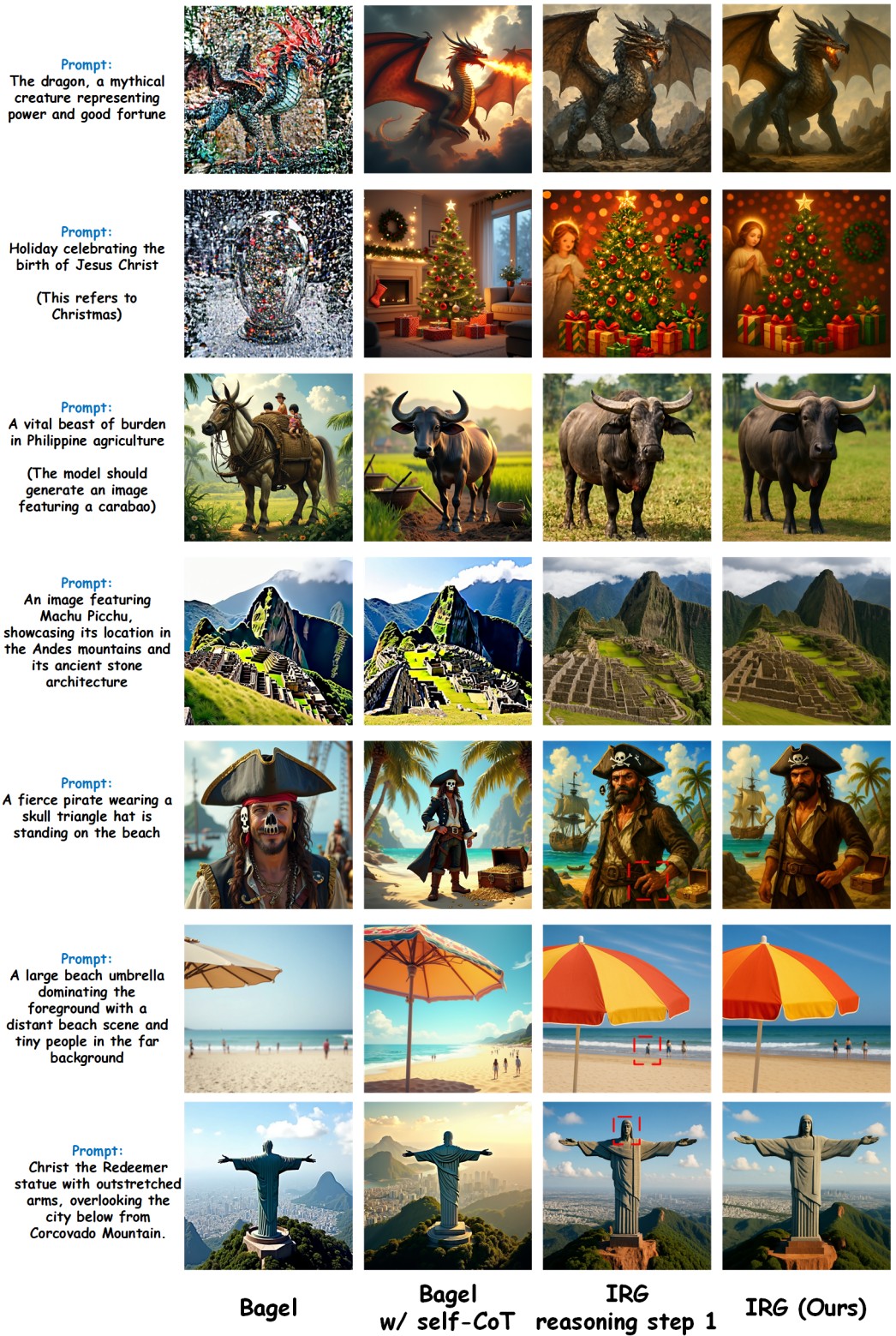

Figure 4: Visualization comparison results of BAGEL (Deng et al., 2025), BAGEL w/ self-CoT (Deng et al., 2025), IRG reasoning step 1 and our proposed IRG at 1024×1024 resolution. The examples are selected from WISE (Niu et al., 2025) and GenAI-Bench (Li et al., 2024a). Red boxes highlight the fine-grained details that have obvious flaws.

Table 5: **Quantitative evaluation results of instruct-following capability on TIIF testmini (QwenVL2.5-72B as the evaluation).** * means we report the reproducing results using the official Github repository and checkpoint. The best Overall and Avg. results are **bolded**.

| Model | Overall | | Basic Following | | | | | | | | Advanced Following | | | | | | | | | | | | Designer | |
|---|---|---|---|---|---|---|---|---|---|---|---|---|---|---|---|---|---|---|---|---|---|---|---|---|
| | | | Avg | | Attribute | | Relation | | Reasoning | | Avg | | Attribute +Relation | | Attribute +Reasoning | | Relation +Reasoning | | Style | | Text | | Real World | |
| | short | long | short | long | short | long | short | long | short | long | short | long | short | long | short | long | short | long | short | long | short | long | short | long |
| FLUX.1-dev (Labs, 2024) | 66.24 | 66.72 | 74.41 | 76.67 | 72.50 | 75.50 | 78.20 | 79.78 | 72.52 | 74.73 | 60.72 | 60.95 | 66.76 | 65.50 | 61.76 | 60.74 | 56.60 | 57.49 | 63.33 | 60.00 | 44.49 | 54.75 | 74.63 | 72.01 |
| FLUX.1-Pro (Labs, 2024) | 63.75 | 63.53 | 71.39 | 73.57 | 70.00 | 68.50 | 68.51 | 79.97 | 75.66 | 72.23 | 64.63 | 61.42 | 70.69 | 72.99 | 62.34 | 57.27 | 64.65 | 57.11 | 63.00 | 63.00 | 34.39 | 36.65 | 69.94 | 66.78 |
| DALL-E 3 (Betker et al., 2023) | 74.47 | 72.94 | 77.35 | 78.40 | 77.62 | 75.00 | 80.22 | 79.67 | 74.22 | 80.54 | 70.11 | 68.45 | 76.65 | 75.05 | 68.39 | 68.07 | 63.64 | 59.92 | 79.31 | 80.00 | 74.07 | 75.51 | 76.12 | 62.69 |
| SD3.5-large (Esser et al., 2024) | 68.69 | 64.92 | 73.72 | 72.10 | 77.50 | 66.50 | 74.79 | 77.16 | 68.87 | 72.64 | 65.59 | 63.41 | 70.85 | 68.22 | 65.03 | 62.93 | 61.03 | 61.66 | 56.67 | 60.00 | 73.30 | 46.15 | 70.15 | 69.03 |
| PixArt-Σ (Chen et al., 2024) | 57.46 | 57.04 | 67.74 | 68.19 | 65.50 | 69.50 | 74.33 | 72.11 | 63.40 | 62.96 | 56.71 | 54.52 | 62.47 | 59.67 | 57.51 | 55.08 | 54.84 | 52.64 | 76.67 | 73.33 | 2.71 | 4.98 | 63.06 | 63.06 |
| Lumina-Next (Zhuo et al., 2024) | 46.83 | 51.81 | 59.62 | 62.48 | 49.50 | 61.50 | 63.30 | 65.51 | 66.04 | 60.44 | 43.72 | 47.20 | 47.52 | 51.35 | 42.65 | 42.06 | 44.90 | 50.87 | 53.33 | 66.67 | 2.71 | 6.33 | 51.49 | 61.57 |
| Hunyuan-DiT (Li et al., 2024c) | 49.14 | 52.67 | 65.39 | 67.79 | 59.00 | 63.00 | 79.89 | 76.82 | 57.27 | 63.56 | 51.61 | 52.25 | 62.49 | 59.93 | 49.14 | 45.71 | 49.38 | 54.74 | 53.33 | 73.33 | 0.45 | 2.26 | 31.34 | 34.70 |
| Show-o (Xie et al., 2024) | 57.34 | 61.33 | 69.99 | 75.30 | 66.50 | 80.00 | 76.47 | 71.88 | 67.00 | 74.04 | 58.25 | 58.19 | 67.21 | 64.33 | 54.26 | 58.86 | 61.38 | 56.19 | 46.67 | 66.67 | 4.98 | 11.31 | 71.64 | 68.66 |
| LightGen (Wu et al., 2025c) | 52.84 | 46.42 | 68.70 | 53.99 | 61.00 | 52.00 | 73.69 | 54.52 | 71.40 | 50.52 | 54.10 | 45.76 | 66.82 | 48.37 | 52.22 | 42.93 | 51.07 | 50.64 | 43.33 | 43.33 | 2.26 | 10.86 | 53.73 | 59.70 |
| SANA 1.5 (Xie et al., 2025a) | 62.57 | 63.48 | 73.92 | 72.31 | 71.50 | 73.00 | 82.21 | 78.39 | 68.04 | 65.52 | 60.36 | 60.36 | 65.65 | 67.33 | 56.41 | 56.13 | 62.20 | 60.18 | 66.67 | 76.67 | 28.51 | 23.53 | 61.94 | 70.52 |
| Infinity (Han et al., 2025) | 60.65 | 59.66 | 70.90 | 71.63 | 73.00 | 73.00 | 73.75 | 74.44 | 65.96 | 67.44 | 59.80 | 57.81 | 68.92 | 63.78 | 60.53 | 56.87 | 55.04 | 56.81 | 56.67 | 73.33 | 22.17 | 26.70 | 69.78 | 61.19 |
| Janus-Pro-7B (Chen et al., 2025c) | 65.38 | 61.10 | 74.99 | 73.19 | 74.50 | 78.00 | 73.69 | 70.51 | 76.77 | 71.04 | 61.77 | 56.03 | 65.71 | 66.48 | 62.01 | 55.62 | 61.16 | 49.34 | 43.33 | 70.00 | 38.46 | 42.08 | 79.48 | 73.51 |
| T2I-R1 (Jiang et al., 2025) | 67.61 | 68.34 | 81.14 | 79.45 | 80.50 | 78.50 | 83.09 | 79.49 | 79.81 | 80.37 | 67.38 | 65.90 | 69.92 | 65.27 | 70.10 | 71.62 | 68.69 | 64.68 | 50.00 | 63.33 | 32.13 | 37.56 | 74.25 | 74.25 |
| BAGEL (Deng et al., 2025)* | 70.97 | 71.79 | 78.16 | 78.12 | 78.00 | 79.50 | 80.24 | 79.08 | 76.25 | 75.77 | 68.23 | 68.19 | 73.37 | 77.49 | 64.36 | 66.15 | 68.92 | 61.48 | 80.00 | 80.00 | 40.72 | 52.40 | 76.87 | 74.63 |
| MidJourney v7 (Midjourney, 2025) | 65.92 | 62.43 | 73.96 | 74.63 | 75.00 | 82.00 | 78.74 | 78.51 | 68.12 | 68.55 | 63.44 | 62.59 | 70.60 | 74.03 | 64.43 | 59.58 | 58.84 | 61.34 | 66.67 | 33.33 | 31.67 | 34.39 | **79.22** | **75.32** |
| Show-o2* (Xie et al., 2025b) | 62.80 | 63.87 | 75.30 | 74.45 | 73.00 | 71.00 | 77.22 | 74.09 | 75.69 | 78.25 | 61.38 | 66.12 | 63.47 | 67.44 | 62.63 | 70.31 | 64.15 | 60.00 | 60.00 | 60.00 | 33.33 | 14.03 | 75.00 | 74.63 |
| BAGEL *w/* self-CoT (Deng et al., 2025)* | 68.06 | 68.78 | 77.63 | 79.40 | 75.00 | 77.00 | 78.55 | 82.37 | 79.33 | 78.81 | 71.24 | 68.20 | 77.65 | 75.37 | 69.77 | 65.87 | 72.93 | 67.91 | 69.93 | 63.33 | 26.24 | 26.70 | 69.78 | 71.64 |
| IRG (Ours) | **76.00** | **73.77** | **83.17** | **81.28** | 81.00 | 76.00 | 82.96 | 81.86 | 85.54 | 85.98 | **75.25** | **74.66** | 75.82 | 77.25 | 78.16 | 77.76 | 73.84 | 72.93 | 90.00 | 70.00 | 43.89 | 47.51 | 72.76 | 74.63 |
| GPT-4o (OpenAI, 2025c) | 84.19 | 84.61 | 85.30 | 86.55 | 81.00 | 82.12 | 86.16 | 84.12 | 88.74 | 94.50 | 81.24 | 79.75 | 81.95 | 81.55 | 80.03 | 79.85 | 80.88 | 75.68 | 76.67 | 86.67 | 92.76 | 90.05 | 89.55 | 88.06 |

Table 6: **GenAI-Bench Evaluation Results.** * means we report the reproducing results using the official Github repository and checkpoint. The best results are **bolded**.

| Model | Basic Prompt | | | | | | Advanced Prompt | | | | | | Overall↑ |
|---|---|---|---|---|---|---|---|---|---|---|---|---|---|
| | Attribute↑ | Scene↑ | Relation | | | Avg↑ | Count↑ | Differ↑ | Compare↑ | Logical | | Avg↑ | |
| | | | Spatial↑ | Action↑ | Part↑ | | | | | Negate↑ | Universal↑ | | |
| SD v2.1 (Rombach et al., 2022) | 0.80 | 0.79 | 0.76 | 0.77 | 0.80 | 0.78 | 0.68 | 0.70 | 0.68 | 0.54 | 0.64 | 0.62 | 0.70 |
| SD-XL (Podell et al., 2024) | 0.84 | 0.84 | 0.82 | 0.83 | 0.89 | 0.83 | 0.71 | 0.73 | 0.69 | 0.50 | 0.66 | 0.63 | 0.73 |
| FLUX.1-dev (Labs, 2024) | 0.87 | 0.88 | 0.87 | 0.85 | 0.87 | 0.87 | 0.75 | 0.78 | 0.74 | 0.45 | 0.70 | 0.64 | 0.76 |
| LWM (Liu et al., 2024) | 0.63 | 0.62 | 0.65 | 0.63 | 0.70 | 0.63 | 0.59 | 0.58 | 0.54 | 0.49 | 0.52 | 0.53 | 0.58 |
| Show-o (Xie et al., 2024) | 0.72 | 0.72 | 0.70 | 0.70 | 0.75 | 0.70 | 0.70 | 0.62 | 0.71 | 0.51 | 0.65 | 0.60 | 0.65 |
| VILA-U (Wu et al., 2024d) | 0.78 | 0.78 | 0.77 | 0.78 | 0.79 | 0.76 | 0.70 | 0.71 | 0.74 | 0.53 | 0.66 | 0.64 | 0.70 |
| Liquid (Wu et al., 2024a) | – | – | – | – | – | – | 0.76 | 0.73 | 0.74 | 0.46 | 0.74 | 0.65 | – |
| UniTok (Ma et al., 2025) | – | – | – | – | – | – | 0.76 | 0.76 | 0.79 | 0.46 | 0.73 | 0.67 | – |
| Mogao-7B (Liao et al., 2025) | – | – | – | – | – | – | 0.77 | 0.74 | 0.77 | 0.53 | 0.71 | 0.68 | – |
| Janus-Pro-7B (Ma et al., 2024) | 0.85 | 0.87 | 0.85 | 0.84 | 0.85 | 0.84 | 0.73 | 0.73 | 0.71 | 0.48 | 0.65 | 0.65 | 0.75 |
| BAGEL* (Deng et al., 2025) | 0.89 | 0.90 | 0.89 | 0.88 | 0.89 | 0.89 | 0.77 | 0.77 | 0.79 | 0.52 | 0.71 | 0.68 | 0.79 |
| T2I-R1 (Jiang et al., 2025) | 0.87 | 0.89 | 0.89 | 0.87 | 0.87 | 0.88 | 0.81 | **0.82** | 0.78 | 0.60 | 0.73 | 0.73 | 0.81 |
| Show-o2* (Xie et al., 2025b) | 0.85 | 0.88 | 0.86 | 0.87 | 0.83 | 0.85 | 0.74 | 0.74 | 0.76 | 0.43 | 0.70 | 0.64 | 0.75 |
| BAGEL *w/* self-CoT* (Deng et al., 2025) | 0.86 | 0.88 | 0.87 | 0.87 | 0.82 | 0.86 | 0.81 | 0.78 | 0.81 | **0.66** | 0.77 | 0.75 | 0.81 |
| IRG (Ours) | **0.90** | **0.92** | **0.91** | **0.90** | **0.90** | **0.90** | **0.84** | 0.78 | **0.83** | **0.66** | **0.80** | **0.77** | **0.84** |

We use Qwen2.5VL (Bai et al., 2025b) to generate the initial thinking and improving thinking processes, while GPT-4o generates high-quality images.

**Training settings.** We adopt the unified multimodal understanding and generation model BAGEL (Deng et al., 2025) as our base model. In the first training stage, we train the base model for 2K steps on the six decomposed learning modes using the cross-entropy (CE) loss and mean squared error (MSE) loss. We then continue training the model for 30K steps on the Initial Full Learning and Improving Full Learning tasks.

## C  DETAILED MAIN RESULTS ANALYSIS

To thoroughly assess the visual generation performance of our model, we conduct evaluation on a series of representative benchmarks that cover complementary aspects of controllable and knowledge-grounded generation. Together, these benchmarks provide a comprehensive view of our model's strengths in alignment, reasoning, stylistic control, and text rendering. In the following, we present detailed comparisons with SoTA baselines and highlight the improvements achieved by our proposed IRG model.

**GenEval.** Tab. 1 reports the quantitative results on the GenEval (Ghosh et al., 2023) benchmark, which evaluates compositional T2I generation across diverse object-centric attributes such as counting, color, and spatial position. We benchmark both generation-only models and unified understanding–generation models. Among generation-only approaches, FLUX.1-dev achieves the

Table 7: **Quantitative evaluation results on OneIG-EN.** The overall score is the average of the five dimensions. The best results are **bolded** and the second best results are underlined.

| Model | Alignment | Text | Reasoning | Style | Diversity | Overall↑ |
|---|---|---|---|---|---|---|
| Janus-Pro (Chen et al., 2025c) | 0.553 | 0.001 | 0.139 | 0.276 | 0.365 | 0.267 |
| BLIP3-o (Chen et al., 2025a) | 0.711 | 0.013 | 0.223 | 0.361 | 0.229 | 0.307 |
| BAGEL (Deng et al., 2025) | 0.769 | 0.244 | 0.173 | 0.367 | 0.251 | 0.361 |
| Show-o2-7B (Xie et al., 2025b) | 0.817 | 0.002 | 0.226 | 0.317 | 0.177 | 0.308 |
| SDv1.5 (Rombach et al., 2022) | 0.565 | 0.010 | 0.207 | 0.383 | **0.429** | 0.319 |
| SDXL (Podell et al., 2024) | 0.688 | 0.029 | 0.237 | 0.332 | 0.296 | 0.316 |
| FLUX.1-dev (Labs, 2024) | 0.786 | **0.523** | **0.253** | 0.368 | 0.238 | **0.434** |
| SANA-1.5 4.8B (PAG) (Xie et al., 2025a) | 0.765 | 0.069 | 0.217 | 0.401 | 0.216 | 0.334 |
| Lumina-Image 2.0 (Qin et al., 2025) | 0.819 | 0.106 | 0.270 | 0.354 | 0.216 | 0.353 |
| BAGEL $w/$ self-CoT (Deng et al., 2025) | 0.793 | 0.020 | 0.206 | 0.390 | 0.209 | 0.324 |
| IRG (Ours) | **0.839** | 0.377 | 0.239 | **0.427** | 0.192 | 0.415 |
| GPT-4o (OpenAI, 2025c) | 0.851 | 0.857 | 0.345 | 0.462 | 0.151 | 0.533 |

best performance with an overall score of 0.82. Within the unified category, our proposed IRG model achieves the best overall score of 0.85, consistently outperforming all baselines across multiple sub-tasks, including challenging aspects such as counting (0.83) and position (0.74). These results establish IRG as a new state of the art on GenEval, demonstrating strong controllability and precise compositional generation capabilities.

**WISE.** Tab. 2 reports the quantitative results on the WISE (Niu et al., 2025) benchmark, which evaluates T2I models on complex semantic understanding and world knowledge reasoning across six domains: culture, time, space, biology, physics, and chemistry. Unlike compositional tests such as GenEval, WISE focuses on knowledge-grounded generation where models must accurately reflect real-world semantics in addition to compositional control. Among generation-only models, FLUX.1-dev achieves the best performance with an overall score of 0.50. Within the unified category, our proposed IRG model establishes a new state of the art with an overall score of 0.77. It consistently outperforms prior unified models across all six domains, achieving 0.78 on cultural knowledge, 0.72 on temporal reasoning, 0.76 on spatial understanding, and above 0.80 in biology and physics. These results indicate that IRG not only improves general controllability but also integrates world knowledge more effectively than existing approaches, setting a new benchmark for semantic alignment in T2I generation.

**TIIF.** Tab. 5 presents the quantitative results on TIIF testmini (Wei et al., 2025), a benchmark specifically designed to evaluate T2I models' ability to interpret and accurately follow complex natural language instructions. The benchmark covers both basic following (attributes, relations, and reasoning) and advanced following (multi-attribute reasoning, compositional control, stylistic adherence, and textual rendering), along with a separate evaluation of designer-oriented prompts. When evaluated with QwenVL2.5-72B as the reference, our IRG model achieves the best overall performance with scores of 76.00 (short) and 73.77 (long). Notably, IRG demonstrates consistent improvements in advanced following tasks, achieving 75.25/74.66 on average, and excelling in compositional cases such as attribute+reasoning (78.16/77.76) and relation+reasoning (73.84/72.93).

Overall, these results highlight the superior instruction-following ability of IRG, which consistently generalizes across different evaluation settings and significantly outperforms prior open-source systems, establishing it as a new state of the art in controllable T2I generation.

**GenAI-Bench.** Tab. 6 reports results on GenAI-Bench (Li et al., 2024a), which probes compositional text-to-visual generation across Basic (Attribute, Scene, and Relation: Spatial/Action/Part) and Advanced (Count, Differ, Compare, Logical: Negate/Universal) prompts. Our model IRG attains the best Overall score 0.84, exceeding strong baselines. IRG delivers the strongest overall performance on GenAI-Bench, combining robust basic grounding (attributes, scenes, relations) with improved compositional and logical generalization on advanced prompts.

**OneIG-EN.** Tab. 7 summarizes the quantitative results on OneIG-EN (Chang et al., 2025), the English track of the OneIG-Bench benchmark that evaluates fine-grained T2I generation along five dimensions: alignment, text rendering, reasoning, style, and diversity. The final overall score is

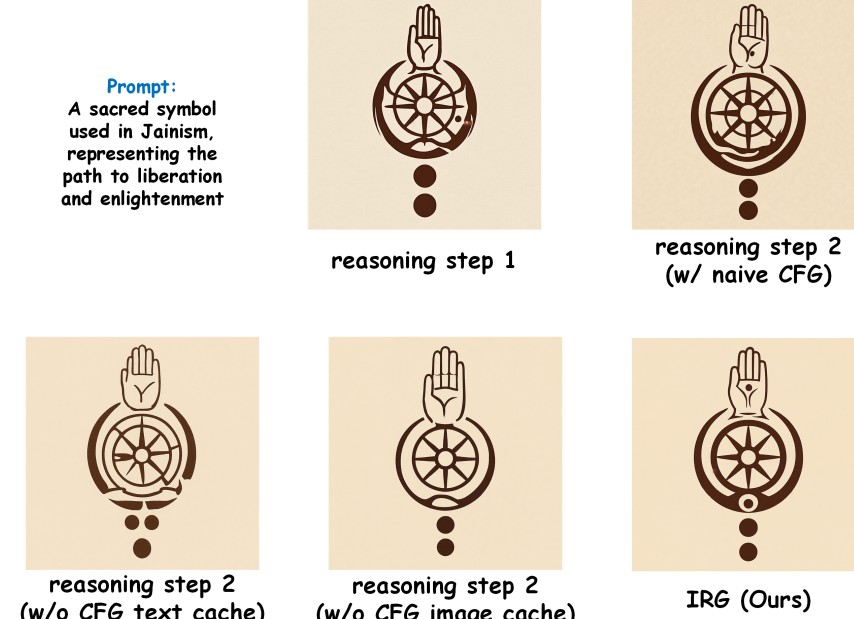

Figure 5: The results of the CFG strategy visualization comparison. "w/ naive CFG" means using the same CFG strategy as in the image generation process of reasoning step 1 (*i.e.*, applying traditional prompt-only CFG at reasoning step 2). "w/o CFG text cache" and "w/o CFG image cache" mean that the tailored CFG text cache and CFG image cache are not used during the improved image generation, respectively. Our proposed IRG uses a dedicated CFG condition design for the improved image generation.

Table 8: **Early stop comparison of IRG.** The base method is BAGEL w/ self-CoT. "IRG w/ early stop" means that we employ the MLLM model to judge whether the IRG reasoning process can be terminated early given the prompt (see Sec. D.5).

| Model | Early stop rate | WISE |
|---|---|---|
| IRG reasoning step1 | - | 0.79 |
| IRG (Ours) | - | 0.77 |
| IRG w/ early stop | 71% | 0.79 |

computed as the average across these dimensions. Our IRG model establishes a new state of the art among open-source approaches with an overall score of (0.415), ranking second only to GPT-4o (0.533). IRG achieves the best alignment (0.839) and style (0.427) scores, and maintains balanced performance across reasoning (0.239) and text rendering (0.377). These results indicate that IRG not only excels in faithfully aligning with user prompts but also produces aesthetically consistent outputs, demonstrating its superior general-purpose generation capability.

# D    MORE DETAILED EXPERIMENTAL RESULTS ANALYSIS

## D.1    VISUALIZATION COMPARISON RESULTS.

As shown in Fig 3, our proposed IRG demonstrates the strong ability to generate the images with high visual quality, fine-grained details, and aesthetic aspects. As illustrated in Fig. 4, compared to BAGEL and BAGEL w/ self-CoT, our proposed IRG achieves superior generation quality and visual fidelity. Moreover, relative to the first-turn generated images, the reflection step in IRG improves the visual quality and fine-grained fidelity of the initially generated images. For example, enhancing suboptimal textures and refining details that were previously poorly rendered. This demonstrates that IRG not only produces images that are semantically correct but also places strong emphasis on fine-detail quality in the generated content.

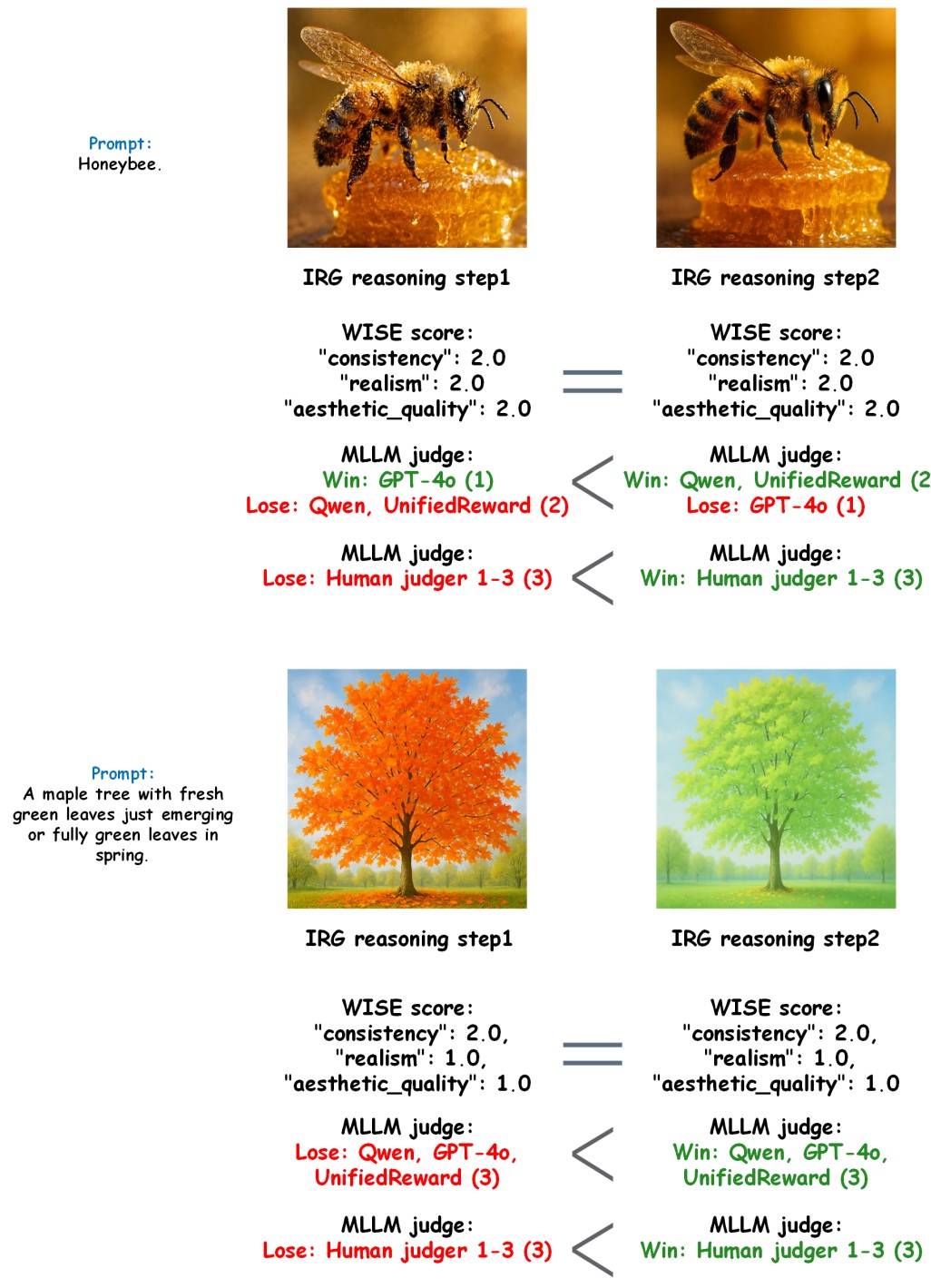

Figure 6: Visualization of the misalignment between benchmark scores and image quality. In this case, IRG reasoning step 2 correctly improves the leaf colors compared to step 1 (where trees had incorrect colors), yet the WISE benchmark score remains identical. This illustrates that standard benchmarks may fail to capture fine-grained visual improvements that are aligned with human and MLLM preferences.

## D.2 CFG STRATEGY.

In Fig. 5, we compare the different CFG strategies in situations that affect improved image generation quality. It is clear that the dedicated CFG condition strategy combining the CFG text condition (conditioning with versus without the initial image information) and the CFG image condition

Prompt:
An artful arrangement of white
buttons outlines a shirt design on a
light-hued fabric, with a contrasting
collar, potentially conveying an artistic
interpretation of a shirt's concept.

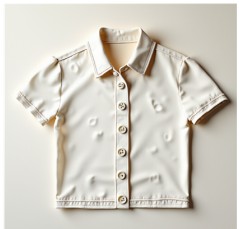

**Original image**

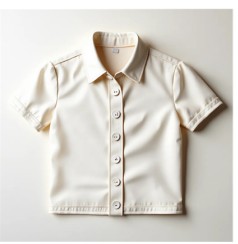 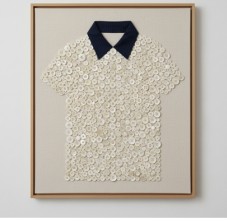 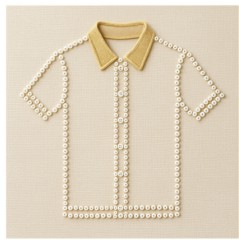

**Qwen-image-edit**  **Hunyuan-image-3.0**  **GPT-4o (ours)**

Figure 7: Qualitative comparison of dataset construction using different foundation models. We compare the refinement capabilities of the open-source Hunyuan-Image-3.0 against GPT-4o. The results indicate that open-source models can effectively perform the reasoning-driven image improvement steps, validating the feasibility of reproducing the dataset construction using open-source resources.

Table 9: **Results of extension to more reasoning steps.**

| Model | WISE | TIIF | GenAI-Bench |
|---|---|---|---|
| IRG reasoning step 1 | 0.79 | 75.84/73.90 | 0.84 |
| IRG reasoning step 2 (Ours) | 0.77 | 76.00/73.77 | 0.84 |
| IRG reasoning step 3 | 0.76 | 74.23/72.96 | 0.83 |
| IRG reasoning step 4 | 0.75 | 74.41/72.81 | 0.83 |

(conditioning with versus without the reflection text) achieves the best visual quality and stability of generated images.

### D.3 RELATION IN BENCHMARK SCORE AND IMAGE QUALITY.

We conducted further analysis, and the results demonstrate that all three MLLMs and three human annotators consistently prefer images generated at IRG reasoning step 2 over step 1. Notably, in the case shown in Fig. 6, IRG reasoning step 1 generated trees with incorrect leaf colors, which were correctly improved in step 2, yet the WISE benchmark scores remained identical. This phenomenon confirms that benchmark scores do not necessarily reflect image quality. The misalignment between text-to-image generation benchmark scores and actual image quality has been a persistent challenge in the field, which the community continues to address. We recognize that standard benchmarks are often insensitive to fine-grained visual improvements and supplement them with multi-MLLM evaluator comparisons, providing convincing qualitative and quantitative validation.

### D.4 OPEN-SOURCE MODEL FOR DATASET CONDUCTION.

We further investigate the feasibility of utilizing open-source models for constructing the IRGL dataset to ensure accessibility and reproducibility. Fig. 7 presents a comparative analysis of cases generated using the open-source model Hunyuan-Image-3.0 (Cao et al., 2025) versus the proprietary GPT-4o. The results demonstrate that the open-source model is equally capable of reasoning and achieving significant, semantically correct improvements over the initial generation. This empirical

Table 10: **More detailed ablation study of IRG.** "- Understanding Learning in Stage1 Training" means that we convert all *Understanding Learning* data to *Generation Learning* and performing Stage 1 training with the same number of steps. "- Full Learning in Stage1 Training" means that we conduct an ablation study removing image learning (*Initial Full Learning* and *Improving Full Learning*) during Stage 1 training.

| Model | WISE | | | | | TIIF | GenAI-Bench |
| --- | --- | --- | --- | --- | --- | --- | --- |
| | Score | Rank score | | | | Score | Score |
| | | Qwen | GPT-4o | UnifiedReward | Avg. | | |
| IRG (Ours) | 0.77 | 53% ↑ | 51% ↑ | 51% ↑ | 51.7% ↑ | 76.00/73.77 | 0.84 |
| - Understanding Learning in Stage1 Training | 0.77 | 47% ↓ | 49% ↓ | 49% ↓ | 48.3% ↓ | 75.34/73.51 | 0.84 |
| - Full Learning in Stage1 Training | 0.56 | - | - | - | - | 55.82/54.31 | 0.72 |

**Prompt:**
Coast redwood tree, highlighting its towering height and dense foliage.

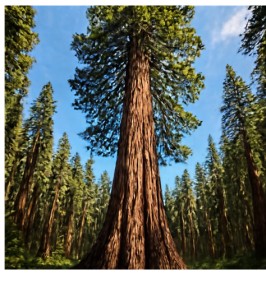 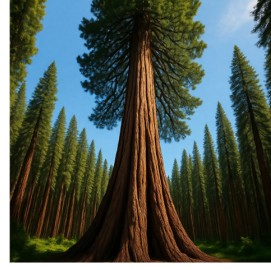

IRG reasoning step1          IRG reasoning step2

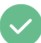 MLLM judge: The generated image in reasoning step 1 is good enough.

**Prompt:**
A maple tree with fresh green leaves just emerging or fully green leaves in spring.

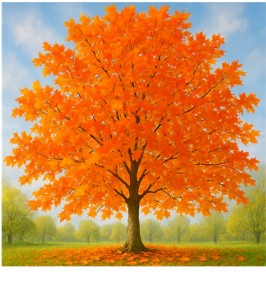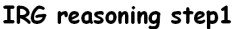 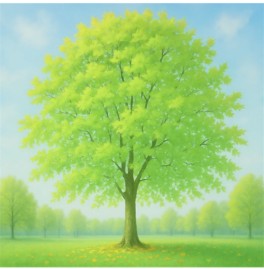

IRG reasoning step1          IRG reasoning step2

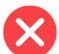 MLLM judge: The generated image in reasoning step 1 is not good.

Figure 8: Visualization of the early stopping phenomenon. The figure illustrates a scenario where the reasoning process is terminated at Step 0, despite the subsequent Step 1 (shown below) demonstrating clear improvements in detail and alignment. This highlights a limitation where the stopping criterion rejects valid refinements.

evidence supports the viability of our data construction pipeline using open-weight models, indicating that the proposed method is not dependent on specific proprietary APIs.

## D.5    EARLY STOP.

Regarding whether generation can be completed at the first stage, we conducted an experimental study on the WISE benchmark. We employed the Qwen3-VL (Bai et al., 2025a) model to judge whether the IRG reasoning process can be terminated early, given the prompt. The results are presented in Tab. 8, introducing an MLLM to judge early termination is feasible (at least without degrading benchmark metrics). However, this introduces a fundamental issue: the MLLM lacks reference information, making it difficult to determine whether subsequent IRG steps will yield further improvements, potentially leading to erroneous judgments.

**Prompt:**
Michael Phieen swimming.

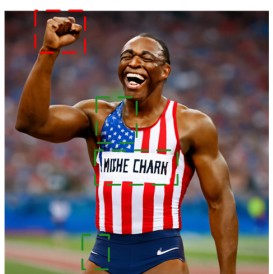
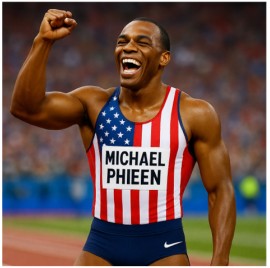

**IRG reasoning step1**          **IRG reasoning step2**

**Prompt:**
The Great Sphinx of Giza
with a lion's body and a
human head, surrounded
by desert.

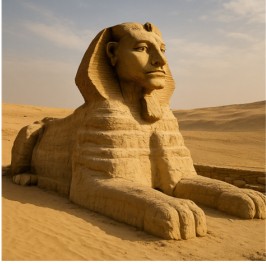
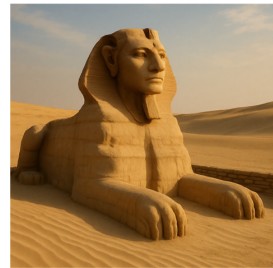

**IRG reasoning step1**          **IRG reasoning step2**

**Prompt:**
A boy is skateboarding on
the Great Wall.

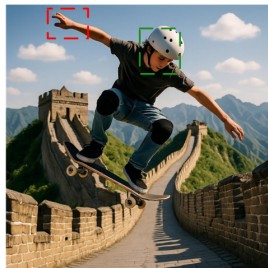
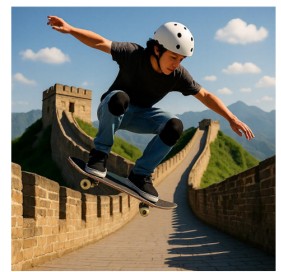

**IRG reasoning step1**          **IRG reasoning step2**

Figure 9: Qualitative analysis of failure cases. This figure illustrates scenarios where the reasoning and refinement process yields suboptimal results. Green boxes denote areas where local improvements were successful, while red boxes indicate regions where the intended corrections failed or introduced new artifacts. These cases highlight the challenges of subjective preference and complex constraint satisfaction.

Table 11: **Inference cost of IRG.**

|                  | IRG reasoning step 1 | IRG reasoning step 2 |
|------------------|:--------------------:|:--------------------:|
| Inference latency | 29.79s              | 60.58s               |
| Peak GPU memory   | 28.23GB             | 29.35GB              |

As shown in Fig. 8,We observe instances where the MLLM-based discriminator triggers an early stop, terminating the inference process prematurely. In these cases, the discriminator halts the reasoning loop even when a subsequent refinement step yields a result with superior visual quality or better prompt alignment than the retained image. This phenomenon suggests that while the current stopping policy effectively prevents degradation, it can occasionally be overly conservative. Enhancing the discriminator's ability to recognize subtle but meaningful improvements remains a promising avenue for future research.

## D.6   ERROR ANALYSIS AND FAILURE MODES.

Visualization results (Fig. 1 and Fig. 4) reveal remaining failure patterns: (1) Micro-structure saturation on repetitive textures (*e.g.*, fabrics, foliage), where the improving step occasionally over-

**Prompt:**
The priestly blessing hand gesture common in Judaism, with fingers spread to form the letter Shin.

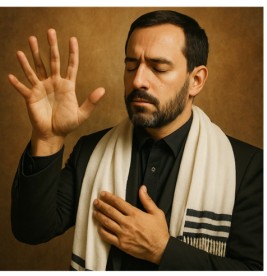

**IRG reasoning step1**

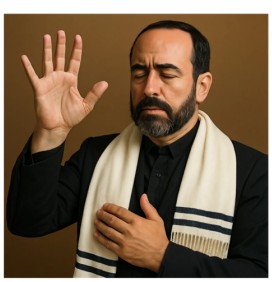

**IRG reasoning step2**

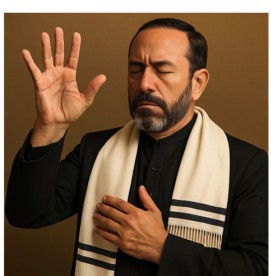

**IRG reasoning step3**

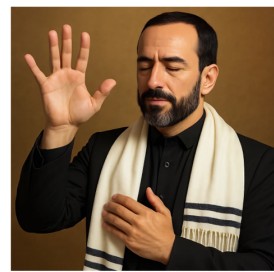

**IRG reasoning step4**

Figure 10: Visualization of extended reasoning steps. The figure displays the progression of generated images across additional iterations (e.g., Step 3 and Step 4). While the model continues to refine the image, the marginal gains decrease after the initial steps, validating the effectiveness of the proposed stopping criteria.

smooths high-frequency details; (2) Text rendering drift under dense constraints, where the refinement trades legibility for stylistic coherence; (3) Global-local tension in crowded scenes, where local edits improve parts while slightly perturbing global layout. We find these are most pronounced when $T_{out}^{(2)}$ introduces many simultaneous edits; a conservative editing policy improves stability but may cap the attainable gains.

We further present a detailed breakdown of failure instances in Fig. 9. In these scenarios, the model attempts to address defects identified in the initial generation (Step 1). While certain regions are successfully rectified (highlighted in green boxes), other intended improvements fail to materialize or are executed incorrectly (highlighted in red boxes). Additionally, we observe instances where the refinement is inherently subjective; in such cases, human evaluators may prefer the initial generation despite the semantic adjustments made during the reasoning steps. Notwithstanding these specific failure modes, our quantitative evaluation confirms that the proposed method yields improvements aligned with human preference in the majority of cases.

## D.7    EXTENDING TO MORE STEPS OF REASONING.

We further investigate the behavior of our framework when extending the inference process to additional reasoning steps beyond the default configuration. As presented in Tab. 9, the results show that extending to more rounds leads to a slight decrease in IRG's benchmark scores. We attribute this to the fact that IRG is trained only on two-round data. Additionally, since IRG already substantially improves image quality by the second round, subsequent improvements are relatively minor, and potential error accumulation may lead to slight performance degradation.

As shown in Fig. 10, the model remains capable of performing valid reasoning and modification in subsequent iterations. However, empirical observations suggest that the most significant quality improvements are typically achieved within the initial refinement steps. Extending the process further tends to yield diminishing returns regarding prompt alignment and visual fidelity, and in some instances, may lead to over-editing. This analysis supports the efficiency of our dynamic stopping mechanism, which balances generation quality with computational cost.

### D.8 MORE DETAILED ABLATION STUDY AND ANALYSIS.

For the more motivation analysis of *Decomposed Learning Modes*, Both *Understanding Learning* and *Generation Learning* are designed to enable the model to learn text-based reasoning capabilities, with key distinctions in their approaches. *Understanding Learning* directly learns text-based reasoning grounded in visual information. For instance, *Initial Thinking Understanding Learning* learns to generate initial thinking given an image, while *Improving Thinking Understanding Learning* learns to produce improving thinking given both first and second round images. In contrast, Generation Learning is designed to directly simulate the actual IRG reasoning process. For example, *Initial Thinking Generation Learning* mimics generating initial thinking directly from the given prompt, while *Improving Thinking Generation Learning* generates improving thinking based on the previously generated image and reasoning chain. This design is motivated by the intuition and empirical observation that learning thinking with visual reference versus learning thinking directly from text poses different learning challenges for the model–the former benefits from contrastive visual references, while the latter requires reasoning with more limited information. Additionally, during Stage 1 training, we observed that these two training approaches exhibit different convergence characteristics.

As shown in Tab. 10, The results demonstrate that employing both *Understanding Learning* and *Generation Learning* during Stage 1 versus using only *Generation Learning* achieves similar benchmark scores. However, in MLLM-based image quality judgments, IRG slightly outperforms the variant without *Understanding Learning*. For the ablation of whether removing image learning during Stage 1 training, this ablation study explicitly demonstrates the necessity of incorporating full thinking-image trajectories during Stage 1 training.

### D.9 INFERENCE COST OF IRG.

As shown in Tab. 11, the two-step reasoning process of IRG does introduce additional inference overhead. However, this overhead primarily manifests in inference latency rather than peak GPU memory consumption (where the majority of GPU memory is occupied by model parameter loading). In this scenario, deploying this model for interleaved reasoning generation in practical applications is generally not constrained, as it does not require substantially additional GPU memory that would limit device deployment. The primary impact is on generation latency, which aligns with the bottleneck observed in LLM reasoning models.

Furthermore, we would like to clarify that this additional overhead is an inherent characteristic of test-time scaling techniques. When we aim to enhance model performance through inference-time techniques, such overhead is expected and unavoidable compared to non-reasoning models, and is not a limitation unique to our approach. Our proposed IRG introduces both text-based and visual-based reasoning into the text-to-image generation domain, representing an innovative contribution to this field and providing a powerful new paradigm for further enhancing generative model capabilities.

### D.10 FULL CASE VISUALIZATION.

We provide comprehensive visualizations of the full inference process in Fig. 11 and 12. These figures display the intermediate reasoning outputs, including the generated images at each step, the corresponding critique and reasoning traces provided by the MLLM, and the refined prompts used for subsequent generation. This transparent view of the interleaving process highlights how the model iteratively identifies specific issues–such as spatial inconsistencies or missing details—and corrects them to align better with the user's intent.

## E    LLM USAGE

In this paper, we have used an LLM to refine some sentences and improve the grammar, making the paper more academic.

**Prompt:**

**A fierce pirate wearing a skull triangle hat is standing on the beach**

<think>
The model should generate an image of a pirate with a skull earring standing on a treasure island, capturing a rugged and adventurous atmosphere with vibrant colors and dynamic lighting that highlight the pirate's weathered features and the allure of the treasure behind him.

Here's the full, detailed prompt: A rugged pirate stands prominently in the foreground, his long, tousled hair and thick beard framing a weathered face marked by years of seafaring adventures. He wears a classic pirate outfit consisting of a tricorn hat adorned with a skull and crossbones, a brown vest over a striped shirt, and a belt with a large buckle. His right hand rests confidently on his hip, while his left hand grips a wooden staff, symbolizing his authority and strength. A distinctive skull earring adds a touch of menace and individuality to his appearance.

The pirate stands on a rocky outcrop overlooking a lush, green treasure island dotted with palm trees and stretches of golden sand. In the background, a majestic pirate ship with multiple masts and billowing sails is anchored near the shore, its sails catching the wind. The ship is detailed with ropes, rigging, and flags, adding to the authenticity of the scene. A small rowboat rests on the sandy beach, ready for use.

The sky above is a vibrant blue, filled with fluffy white clouds that cast soft shadows over the landscape, creating a sense of depth and dimension. The lighting is bright and natural, highlighting the textures of the pirate's clothing, the rocks beneath his feet, and the gleaming treasure chest filled with gold coins. The overall mood is one of excitement and adventure, with a hint of danger and mystery. The color palette is rich and warm, with earthy tones dominating the foreground and cooler blues and greens in the background, enhancing the contrast and visual appeal of the scene. The composition is balanced, with the pirate as the central focus, drawing the viewer's eye towards him and then outwards to the treasure and the ship, creating a narrative of exploration and discovery.
</think>

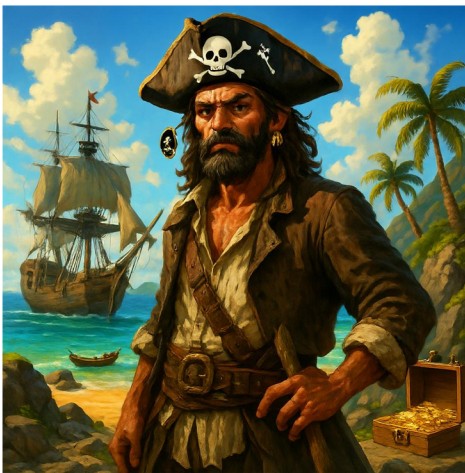

**IRG reasoning step 1**

<think>
The previous generated image successfully captures the essence of a pirate on a treasure island but falls short in several areas. The pirate's expression appears somewhat neutral rather than conveying the rugged determination typical of a pirate. Additionally, the background elements like the treasure chest and the pirate ship lack detail and vibrancy, which detracts from the overall adventurous atmosphere. The lighting and color palette could also be enhanced to create a more dynamic and immersive scene.

### Detailed Explanation of Required Improvements:

The pirate's facial expression needs to be more intense and commanding, reflecting his role as a fearless leader. This can be achieved by adding a slight frown and deepening the lines around his eyes and mouth. The pirate ship in the background should be more detailed, with visible rigging, sails, and flags, enhancing its realism. The treasure chest should be more prominent and detailed, with gold coins spilling out, adding to the allure of the scene. The lighting should be adjusted to create a more dramatic effect, with shadows and highlights emphasizing the textures of the pirate's clothing and the surrounding environment. The color palette should be richer and more vibrant, with deeper blues and greens in the sky and sea, and warmer tones in the foreground.

### Step-by-Step Modification Guidance:

1. Adjust the pirate's facial expression to be more intense and commanding, with a slight frown and deepened lines around his eyes and mouth.
2. Enhance the details of the pirate ship in the background, including rigging, sails, and flags, to make it more realistic.
3. Make the treasure chest more prominent and detailed, with gold coins spilling out, adding to the scene's allure.
4. Modify the lighting to create a more dramatic effect, using shadows and highlights to emphasize the textures of the pirate's clothing and the surrounding environment.
5. Adjust the color palette to be richer and more vibrant, with deeper blues and greens in the sky and sea, and warmer tones in the foreground.

### Final Comprehensive Prompt for the Improved Image:

A rugged pirate stands prominently in the foreground, his long, tousled hair and thick beard framing a determined face marked by years of seafaring adventures. He wears a classic pirate outfit consisting of a tricorn hat adorned with a skull and crossbones, a brown vest over a striped shirt, and a belt with a large buckle. His right hand rests confidently on his hip, while his left hand grips a wooden staff, symbolizing his authority and strength. A distinctive skull earring adds a touch of menace and individuality to his appearance.

The pirate stands on a rocky outcrop overlooking a lush, green treasure island dotted with palm trees and stretches of golden sand. In the background, a majestic pirate ship with multiple masts and billowing sails is anchored near the shore, its sails catching the wind. The ship is detailed with ropes, rigging, and flags, adding to the authenticity of the scene. A small rowboat rests on the sandy beach, ready for use.

The sky above is a vibrant blue, filled with fluffy white clouds that cast soft shadows over the landscape, creating a sense of depth and dimension. The lighting is bright and natural, highlighting the textures of the pirate's clothing, the rocks beneath his feet, and the gleaming treasure chest filled with gold coins. The overall mood is one of excitement and adventure, with a hint of danger and mystery. The color palette is rich and warm, with earthy tones dominating the foreground and cooler blues and greens in the background, enhancing the contrast and visual appeal of the scene. The composition is balanced, with the pirate as the central focus, drawing the viewer's eye towards him and then outwards to the treasure and the ship, creating a narrative of exploration and discovery.
</think>

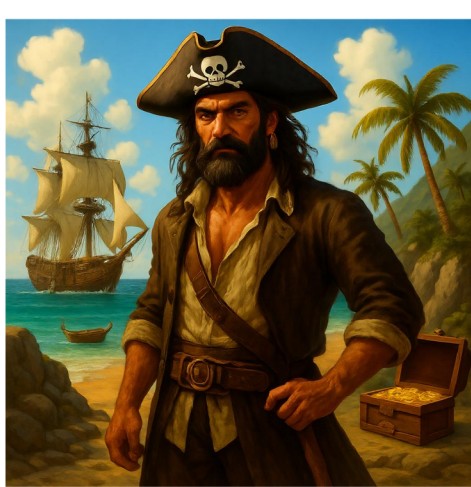

**IRG reasoning step 2**

Figure 11: Full case visualization (Part I). This figure illustrates the complete reasoning chain for selected samples. It shows the input prompt, the initial image generation, the specific reasoning feedback (critique) generated by the model, and the resulting improved image. This demonstrates the model's ability to self-correct based on its own reasoning.

**Prompt:**
**Easter egg hunt**

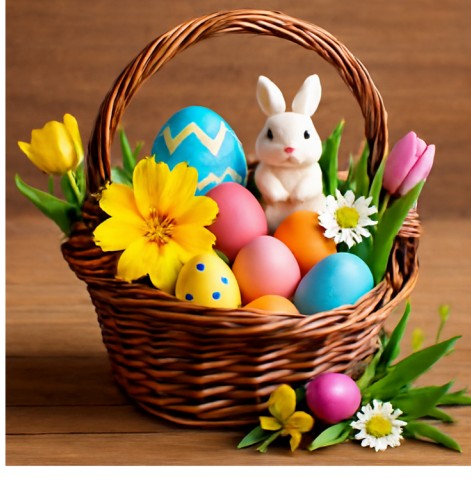

**IRG reasoning step 1**

<think>
The model should generate an image of a traditional Easter basket filled with colorful chocolate eggs, decorated with a vibrant spring flower bouquet and a yellow egg, set against a warm wooden background to evoke a festive and cheerful Easter atmosphere.
Here's the full, detailed prompt: Create an image of a classic Easter basket, woven from dark brown wicker with a sturdy handle, brimming with an assortment of colorful chocolate eggs in shades of yellow, orange, pink, blue, and purple. The eggs are arranged neatly within the basket, their smooth surfaces reflecting light subtly. Adorning the top of the basket is a charming yellow egg with a blue zigzag pattern, sitting beside a small, fluffy white bunny figurine. Beside the basket, a delicate spring flower bouquet consisting of pink tulips, white daisies, and yellow blossoms adds a touch of freshness and vibrancy. The flowers are accompanied by lush green leaves, enhancing the natural and festive feel. The entire scene is set against a warm, smooth wooden surface that provides a soft contrast to the bright colors of the eggs and flowers. The lighting is soft and natural, casting gentle shadows and highlighting the textures of the basket, the glossy eggs, and the petals. The overall composition is balanced and inviting, evoking a sense of joy, warmth, and celebration associated with traditional Easter festivities. The color palette is bright and cheerful, with a harmonious blend of pastel and vivid hues that enhance the festive mood.
</think>

<think>
The previous generated image successfully captures the essence of a traditional Easter basket but falls short in several areas. The arrangement of the eggs appears somewhat cluttered, and the flower bouquet lacks vibrancy and variety. Additionally, the lighting seems flat, failing to fully highlight the textures and colors of the objects. The overall composition could benefit from a more balanced and visually appealing arrangement.

### Detailed Explanation of Required Improvements:

To elevate the image, we need to focus on refining the composition, enhancing the color palette, and improving the lighting. The eggs should be arranged more deliberately to create a pleasing visual flow, ensuring that no single color dominates the scene. The flower bouquet requires additional variety and depth, incorporating different types of flowers and leaves to add texture and interest. The lighting should be adjusted to cast more dynamic shadows and highlights, emphasizing the three-dimensional quality of the objects. Lastly, the overall color balance should be fine-tuned to achieve a more harmonious and vibrant look.

### Step-by-Step Modification Guidance:

1. **Composition Adjustment**: Reorganize the eggs within the basket to create a more balanced and aesthetically pleasing arrangement. Ensure that the eggs are not overcrowded and that there is enough space for the flowers and the bunny figurine. Place the eggs in a way that allows their colors to complement each other naturally.

2. **Flower Bouquet Enhancement**: Introduce a wider variety of flowers and leaves to the bouquet. Add more types of flowers, such as daisies or forget-me-nots, along with some green foliage to provide contrast and depth. Position the flowers strategically around the basket to frame the eggs and the bunny effectively.

3. **Lighting Improvement**: Adjust the lighting to create a more dynamic and natural effect. Use a soft, diffused light source to cast gentle shadows and highlights on the eggs, flowers, and basket. This will help to bring out the textures and colors more vividly and give the scene a more lifelike appearance.

4. **Color Balance Refinement**: Fine-tune the color balance to achieve a more harmonious and vibrant look. Ensure that the colors of the eggs, flowers, and basket blend together seamlessly without any one color overpowering the others. Make sure the colors are bright and true to life, contributing to the festive and cheerful atmosphere.

### Final Comprehensive Prompt for the Improved Image:

Create an image of a classic Easter basket, woven from dark brown wicker with a sturdy handle, brimming with an assortment of colorful chocolate eggs in shades of yellow, orange, pink, blue, and purple. Arrange the eggs in a balanced and aesthetically pleasing manner, ensuring they are not overcrowded and have enough space for the flowers and the bunny figurine. Adorn the top of the basket with a charming yellow egg featuring a blue zigzag pattern and a small, fluffy white bunny figurine. Beside the basket, place a vibrant and varied flower bouquet consisting of pink tulips, white daisies, yellow blossoms, and lush green leaves. Set the scene against a warm, smooth wooden surface that provides a soft contrast to the bright colors. Use soft, diffused lighting to cast gentle shadows and highlights, emphasizing the textures and colors of the objects. Fine-tune the color balance to achieve a harmonious and vibrant look, ensuring the colors are bright and true to life. The overall composition should be balanced and inviting, evoking a sense of joy, warmth, and celebration associated with traditional Easter festivities.
</think>

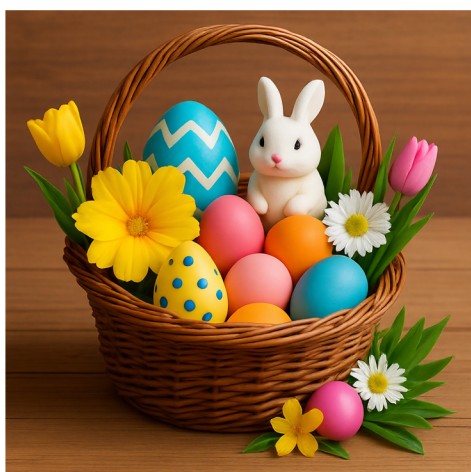

**IRG reasoning step 2**

Figure 12: Full case visualization (Part II). Additional examples of the iterative refinement process. The visualization highlights how the interleaving reasoning mechanism effectively handles complex prompts by breaking down the generation into manageable refinement steps, ensuring high fidelity to the text description.

