# OpenReview forum: "Interleaving Reasoning for Better Text-to-Image Generation"
_ICLR.cc/2026/Conference — ICLR 2026 Poster_

### Official Review · Reviewer_PpTp · 2025-10-17

**Soundness:** 2
**Presentation:** 3
**Contribution:** 4
**Rating:** 4
**Confidence:** 4

**Summary:**

The paper introduces a novel **Interleaving Reasoning Generation (IRG)** framework for text-to-image (T2I) generation. The key idea is to alternate between text-based reasoning and image synthesis, enabling the model to progressively refine image quality and semantics. To support this, the authors curate a **large-scale IRGL-300K dataset** comprising six decomposed learning modes designed to teach both textual reasoning and full image generation trajectories. Experimental results on **5** benchmarks, including **GenEval, WISE, TIIF, GenAI-Bench, and OneIG-EN**, show consistent improvements over existing T2I models or reasoning T2I approaches.

**Strengths:**

1. The paper is well-structured and clearly presented, with thoughtful visual aids such as color-coded pipeline diagrams (e.g., Fig. 2) that make complex processes easy to follow.
2. It addresses an **important and timely challenge** in multimodal generation to bridge reasoning and image synthesis, and provides not only a methodological contribution but also a valuable 300k-sample dataset, which could benefit future research.
3. The experimental validation is strong, demonstrating clear and consistent gains across multiple benchmarks, indicating that the proposed approach is both effective and generalizable.

**Weaknesses:**

1. **Lack of ablation on pipeline design rationale**. While the paper offers a detailed description of the six learning modes and two-stage training, it does not empirically justify whether each component is necessary. For example, it remains unclear if *Initial Thinking Understanding Learning* is crucial, or if training could skip directly to *Initial Thinking Generation Learning* since in Eq. 3, $T_{in}$ will not consider any visual input to generate $T\_{out}^{(1)}$. An ablation isolating these stages would help clarify the necessity and impact of each design choice.
2. **Limited qualitative demonstrations**.  The paper would benefit from showing more examples of intermediate reasoning outputs (e.g., textual thoughts and intermediate images). Figure 4 showcases final results, but visualizing reasoning steps, similar to the teaser in Fig. 1, would provide clearer insight into how the reasoning process contributes to image refinement.
3. **Insufficient comparison to closely related works**. The paper overlooks recent open-source approaches with similar reasoning-generation mechanisms, such as *Self-Correcting with LLM* [1] and *CoT-based Image Generation* [2]. Including these in the comparison, even qualitatively or in terms of design philosophy, would strengthen the paper’s contextual positioning.
4. **Limited exploration of higher reasoning steps**. The paper only discusses cases where $n \le 2$. Extending the study to higher reasoning depths could further emphasize the strength of the proposed *interleaving* reasoning design. For instance, even if trained with $n=2$, the model could use a sliding window of size 2 to perform continuous reasoning, achieving $ n=3, 4$, or even $5$. This would provide deeper insights into how well the model generalizes to multi-step reasoning scenarios.

---

**Overall:**
This is a promising and well-written paper tackling an important topic in T2I reasoning. The contributions are solid, but the lack of discussion and empirical validation of the training pipeline design slightly weakens the argument for its necessity. The current score is 4, which could be raised if the paper includes clearer ablations and more intermediate reasoning visualizations.

---

[1] Wu, T. H. et al. *Self-Correcting LLM-Controlled Diffusion Models.* CVPR 2024.
[2] Guo, Z. et al. *Can We Generate Images with CoT? Let's Verify and Reinforce Image Generation Step by Step.* CVPR 2025.

**Questions:**

1. Why can the proposed training pipeline validate the process? (weakness 1)
2. Can the model generalize to a higher reasoning step with a fixed window size? (weakness 4)
3. Given Eq. 4, what will the $T_q$ look like?
4. What is the inference time comparison compared to other approaches? Since this approach requires extra reasoning, will this double the inference time?
---
 **(suggestion)**: It can increase the readability to add a prefix in Eq. 1. For example, put "image understanding" for the first line and "T2I with CoT" for the second line.

---

> ### Author Response · Authors · 2025-12-03
> **Rebuttal (1/2)**
>
> > W1: More Detailed Ablation Studies
>
> We appreciate your suggestion.
>
> Both *Understanding Learning* and *Generation Learning* are designed to enable the model to learn text-based reasoning capabilities, with key distinctions in their approaches. *Understanding Learning* directly learns text-based reasoning grounded in visual information. For instance, *Initial Thinking Understanding Learning* learns to generate initial thinking given an image, while *Improving Thinking Understanding Learning* learns to produce improving thinking given both first and second round images. In contrast, *Generation Learning* is designed to directly simulate the actual IRG reasoning process. For example, *Initial Thinking Generation Learning* mimics generating initial thinking directly from the given prompt, while *Improving Thinking Generation Learning* generates improving thinking based on the previously generated image and reasoning chain. This design is motivated by the intuition and empirical observation that learning thinking with visual reference versus learning thinking directly from text poses different learning challenges for the model—the former benefits from contrastive visual references, while the latter requires reasoning with more limited information. Additionally, during Stage 1 training, we observed that these two training approaches exhibit different convergence characteristics.
>
> We conducted an experiment converting all *Understanding Learning* data to *Generation Learning* and performing Stage 1 training with the same number of steps. We found that this approach results in slower CE loss convergence, with the final CE loss being slightly higher than the original IRG Stage 1 training. We compared the performance differences between these two Stage 1 training approaches, as shown below:
>
> |                                     | WISE  |      |        |               | TIIF        | GenAI-Bench |
> | ----------------------------------- | ----- | ---- | ------ | ------------- | ----------- | ----------- |
> |                                     | Score | Qwen | GPT-4o | UnifiedReward | Score       | Score       |
> | IRG remove *Understanding Learning* | 0.77  | 47%  | 49%    | 49%           | 75.34/73.51 | 0.84        |
> | IRG (ours)                          | 0.77  | 53%  | 51%    | 51%           | 76.00/73.77 | 0.84        |
>
> The results demonstrate that employing both *Understanding Learning* and *Generation Learning* during Stage 1 versus using only *Generation Learning* achieves similar benchmark scores. However, in MLLM-based image quality judgments, IRG slightly outperforms the variant without *Understanding Learning*.
>
> Furthermore, following the suggestion in Reviewer 5XBw's W1, we conducted an ablation study removing image learning (*Initial Full Learning* and *Improving Full Learning*) during Stage 1 training, with results presented below:
>
> |                                           | WISE | TIIF        | GenAI-Bench |
> | ----------------------------------------- | ---- | ----------- | ----------- |
> | IRG w/o Image Learning in Stage1 Training | 0.56 | 55.82/54.31 | 0.72        |
> | IRG (ours)                                | 0.77 | 76.00/73.77 | 0.84        |
>
> This ablation study explicitly demonstrates the necessity of incorporating full thinking-image trajectories during Stage 1 training.
>
> > W2: More Examples of Intermediate Reasoning Outputs
>
> We appreciate your suggestion.
>
> We have added examples of intermediate reasoning outputs in Figures 11 and 12.
>
> > W3: Comparison with Related Works
>
> We have added discussions of related work on *Self-Correcting with LLM* and *CoT-based Image Generation* in Appendix A. The core distinctions between our proposed IRG and these approaches are as follows:
>
> - *Self-Correcting with LLM* and *CoT-based Image Generation* typically introduce an external reward model or LLM for judgment to guide reflective image generation. In contrast, our proposed IRG directly integrates this reflection process into an end-to-end framework.
> - *Self-Correcting with LLM* and *CoT-based Image Generation* generally focus solely on image-based reflection, whereas IRG proposes a novel "text-image-text-image" interleaved reasoning framework that aims to guide image generation improvements through text-based reasoning.
> - *Self-Correcting with LLM* and *CoT-based Image Generation* are typically constructed for specific benchmarks (e.g., GenEval) and focus on semantic and content correctness, often neglecting the actual quality of generated images. In contrast, IRG demonstrates that through broad paradigm learning, significant improvements can be achieved across multiple diverse benchmarks and in visual quality.

---

> > ### Author Response · Authors · 2025-12-03
> > **Rebuttal (2/2)**
> >
> > > W4: Extension to More Reasoning Steps
> >
> > We greatly appreciate your valuable suggestion.
> >
> > We provide experimental results with additional reasoning steps, evaluating 3-round and 4-round generation. We continue the reasoning process based on previous rounds, with the CFG design only considering the prompt and current round context (i.e., CFG information utilizes only the most recent round's generated image and text reasoning). The results are presented below:
> >
> > |                             | WISE | TIIF        | GenAI-Bench |
> > | --------------------------- | ---- | ----------- | ----------- |
> > | IRG reasoning step 1        | 0.79 | 75.84/73.90 | 0.84        |
> > | IRG reasoning step 2 (ours) | 0.77 | 76.00/73.77 | 0.84        |
> > | IRG reasoning step 3        | 0.76 | 74.23/72.96 | 0.83        |
> > | IRG reasoning step 4        | 0.75 | 74.41/72.81 | 0.83        |
> >
> > The results show that extending to more rounds leads to a slight decrease in IRG's benchmark scores. We attribute this to the fact that IRG is trained only on two-round data. Additionally, since IRG already substantially improves image quality by the second round, subsequent improvements are relatively minor, and potential error accumulation may lead to slight performance degradation.
> >
> > Furthermore, we provide visualization of 4-round reasoning in Fig. 10. As shown, the model has already improved most details of the step 1 generated images, and subsequent improvements are not particularly significant. However, there is no evidence of severe generation collapse, demonstrating that our IRG, despite being trained only on two-round data, still exhibits generalization capability to longer reasoning sequences.
> >
> > > Q1: Validation of Training Pipeline
> >
> > Please refer to our response to W1.
> >
> > > Q2: High Reasoning Steps
> >
> > Please refer to our response to W4.
> >
> > > Q3: $T_q$
> >
> > $T_q$ in Eq. 4 is a specifically designed question intended to enable the model to learn how to generate the initial thinking for an image given that image. The actual prompt we use in our experiments is as follows:
> >
> > ```
> > You have been given one prompt and one image. You should think about the
> > planning process for how to generate this image in the mind.
> >
> > The planning process is enclosed within <think> </think> tags, i.e.
> > <think> planning process here
> > ```
> >
> > > Q4: Inference Time
> >
> > We appreciate your suggestion and have supplemented information regarding inference efficiency.
> >
> > We measured inference latency and peak GPU memory consumption on a single H800 GPU, averaged over 100 prompts. The results are presented below:
> >
> > |                    | IRG reasoning step 1 | IRG reasoning step 2 |
> > | ------------------ | -------------------- | -------------------- |
> > | Inference latency  | 29.79s               | 60.58s               |
> > | Peak GPU memory    | 28.23GB              | 29.35GB              |
> >
> > As shown in the table, the two-step reasoning process of IRG does introduce additional inference overhead. However, this overhead primarily manifests in inference latency rather than peak GPU memory consumption (where the majority of GPU memory is occupied by model parameter loading). Consequently, deploying this model for interleaved reasoning generation in practical scenarios is generally not constrained, as it does not require substantially additional GPU memory that would limit device deployment. The primary impact is on generation latency, which is consistent with the bottleneck observed in LLM reasoning models.
> >
> > Furthermore, we would like to clarify that this additional overhead is an inherent characteristic of test-time scaling techniques. When we aim to enhance model performance through inference-time reasoning, such overhead is expected and unavoidable compared to non-reasoning models, and is not a limitation unique to our approach. Our proposed IRG introduces both text-based and visual-based reasoning into the text-to-image generation domain, representing an innovative contribution that provides a powerful new paradigm for further enhancing generative model capabilities.

---

### Official Review · Reviewer_bEWK · 2025-10-30

**Soundness:** 3
**Presentation:** 3
**Contribution:** 3
**Rating:** 4
**Confidence:** 3

**Summary:**

This paper proposes Interleaving Reasoning Generation (IRG), a framework that alternates between text-based thinking and image synthesis to improve T2I generation quality. The key idea is: (1) generate initial thinking and an initial image, (2) reflect on the initial image to identify improvements, and (3) generate a refined image. To train IRG, the authors propose IRGL-300K, a dataset with six decomposed learning modes covering text-based thinking and full thinking-image trajectories. A two-stage training pipeline is employed: Stage 1 builds reasoning and reflection capabilities across all six tasks, while Stage 2 fine-tunes the complete IRG pipeline. Experiments show improvements of 5-10 points on GenEval, WISE, TIIF, GenAI-Bench, and OneIG-EN benchmarks.

**Strengths:**

**Intuitive Framework**: The paper's core contribution, the Interleaving Reasoning Generation (IRG) framework, is conceptually elegant and highly intuitive. The "think-generate-reflect-refine" process mirrors human creative strategies, making the approach easy to understand and appreciate.

**Methodological Depth**: The creation of the IRGL-300K dataset and the design of the two-stage, six-mode training process demonstrate a deep, systematic approach to solving a challenging training problem.

**Weaknesses:**

The paper presents an interesting idea but is undermined by significant methodological flaws, an absence of critical analysis, and unconvincing experimental results. The core weaknesses fall into two main categories: the questionable contribution of the proposed method and the unaddressed practical costs.

**Unjustified Complexity and Missing Cost-Benefit Analysis**

The primary weakness is the paper's failure to justify the substantial increase in complexity and cost. The Interleaving Reasoning Generation (IRG) framework introduces significant overhead at both training and inference stages, yet the benefits are not clearly demonstrated.
Critically Absent Efficiency Analysis: The proposed 2-turn method involves a 5-step sequential pipeline (initial thought → initial image → encode → reflection → refined image). This inherently introduces significant latency compared to single-step generation. The paper completely omits any discussion of inference efficiency, including critical metrics like inference time, memory consumption, or throughput. Without this analysis, it is impossible to assess the practical viability of the method.


**Questionable Marginal Utility**

The paper's own results cast doubt on the core value proposition of the second "improving" turn.
Evidence of Negative Returns: According to Table 4, single-turn generation from the IRG-trained model outperforms the 2-turn generation on the WISE benchmark (0.79 vs 0.77). This suggests that the reflection and refinement step can actually degrade image quality. The paper fails to investigate or explain these failure cases, leaving a critical gap in understanding the method's behavior.

**Questions:**

The paper's core idea is intuitive, but the proposed solution is exceptionally heavyweight, involving significant costs for both data creation and inference. Given this high investment, the actual benefits need to be clearly demonstrated, but the evidence presented is unconvincing and contradictory. Specifically, the results in Table 4 are confusing, as automated benchmarks show little to no improvement from the second reasoning turn. Furthermore, in the qualitative examples (Figure 4), the refined image can lose important details. The central issue is whether the method's substantial costs are justified by a reliable and significant improvement in image quality. The current results fail to make a convincing case. I hope the authors can provide clarification on these critical points.

**Training & Data Cost**: What were the resource costs (e.g., API expenses, GPU hours) to create the IRGL-300K dataset and train the model?

**Inference Cost**: How much slower is the 2-turn generation process compared to a single turn? Can you provide concrete numbers for inference time and memory usage?

**Evaluation Contradiction**: MLLM ranking prefers 2-turn generation, but benchmark scores in Table 4 do not. Have you conducted human evaluations to confirm that the second turn offers a genuine improvement and rule out potential MLLM bias?

---

> ### Author Response · Authors · 2025-12-03
> **Rebuttal (1/3)**
>
> > W1: Inference Cost
>
> We appreciate your suggestion and have supplemented information regarding inference efficiency.
>
> We measured inference latency and peak GPU memory consumption on a single H800 GPU, averaged over 100 prompts. The results are presented below:
>
> |                    | IRG reasoning step 1 | IRG reasoning step 2 |
> | ------------------ | -------------------- | -------------------- |
> | Inference latency  | 29.79s               | 60.58s               |
> | Peak GPU memory    | 28.23GB              | 29.35GB              |
>
> As shown in the table, the two-step reasoning process of IRG does introduce additional inference overhead. However, this overhead primarily manifests in inference latency rather than peak GPU memory consumption (where the majority of GPU memory is occupied by model parameter loading). Consequently, deploying this model for interleaved reasoning generation in practical scenarios is generally not constrained, as it does not require substantially additional GPU memory that would limit device deployment. The primary impact is on generation latency, which is consistent with the bottleneck observed in LLM reasoning models.
>
> Furthermore, we would like to clarify that this additional overhead is an inherent characteristic of test-time scaling techniques. When we aim to enhance model performance through inference-time reasoning, such overhead is expected and unavoidable compared to non-reasoning models, and is not a limitation unique to our approach. Our proposed IRG introduces both text-based and visual-based reasoning into the text-to-image generation domain, representing an innovative contribution that provides a powerful new paradigm for further enhancing generative model capabilities.
>
> > W2-1: Actual Benefits of IRG
>
> We appreciate your suggestion, but we respectfully disagree with your assessment.
>
> **First, we do not agree that when IRG's WISE benchmark score is lower than IRG reasoning step 1, "This suggests that the reflection and refinement step can actually degrade image quality."**
>
> We must emphasize that in the text-to-image generation domain, most benchmark scores do not directly represent "image quality." Rather, they typically struggle to capture the actual quality, fine-grained details, and aesthetic aspects of images. For instance, the WISE benchmark you mentioned primarily evaluates the model's world knowledge and corresponding generation capabilities. WISE benchmark uses GPT-4o to judge whether generated images conform to specified entities and assigns corresponding scores. However, this evaluation is limited by LLM's inherent scoring mechanism (lacking comparative references to distinguish image quality and fine-grained details [1]), making it difficult to differentiate the actual quality of images generated by IRG reasoning steps 1 and 2 (as shown in Lines 460-477 and Table 4, which is why we introduce MLLM-based direct ranking between IRG steps 1 and 2).
>
> We have supplemented this phenomenon with analysis in Fig. 6 and Appendix D.3. We evaluated images generated by IRG's two steps using three different MLLMs and human annotators (details regarding human annotators can be found in our response to W2 of Reviewer JwV8). The results demonstrate that all three MLLMs and three human annotators consistently prefer images generated at IRG reasoning step 2 over step 1. Notably, in Case 2, IRG reasoning step 1 generated trees with incorrect leaf colors, which were correctly improved in step 2, yet the WISE benchmark scores remained identical. This phenomenon confirms that benchmark scores do not necessarily reflect "image quality."
>
> The misalignment between text-to-image generation benchmark scores and actual image quality has been a persistent challenge in the field, which the community continues to address. This is why numerous works rely on visualization and image comparisons to validate generation quality (we similarly provide extensive comparisons in Fig. 4). As Reviewer 5XBw noted, "The authors recognize that standard benchmarks are often insensitive to fine-grained visual improvements and supplement them with multi-MLLM evaluator comparisons, providing convincing qualitative and quantitative validation." We believe this is a fundamental understanding within the community.

---

> > ### Author Response · Authors · 2025-12-03
> > **Rebuttal (2/3)**
> >
> > **Second, we wish to clarify that the goal of IRG's multi-turn reasoning is not merely to achieve higher benchmark scores at reasoning step 2 compared to step 1, but rather to further enhance the quality, fine-grained details, and aesthetic aspects building upon step 1.**
> >
> > As stated in Lines 120-122, IRG's objective is: "we not only enhance the semantic correctness of the generated content, but also focus on improving the quality, fine-grained details, and aesthetic aspects of the generated images." This motivation also drives the design of our corresponding IRGL. We adopt full IRG trajectories that emphasize generation quality improvement rather than solely fitting benchmarks (although we still significantly improve benchmark performance). This approach enables us to substantially enhance both the model's generation capabilities and image quality, which sufficiently demonstrates the effectiveness of our proposed method.
> >
> > **Third, we emphasize that the performance gains from IRGL's specialized optimization objective during training remain significant.**
> >
> > As shown in Table 3, our proposed IRGL significantly improves generation performance compared to directly training with high-quality images. This benefits from IRGL's transformation of the difficulty of directly learning to generate high-quality images into learning the details needed to improve from a suboptimal self-generated image to a high-quality image. This learning paradigm enables the model to better capture the steps and conditions required for high-quality image generation. Consequently, even at IRG reasoning step 1, the model achieves superior image generation capabilities compared to directly learning from high-quality images. This represents an additional advantage and contribution of our proposed IRGL.
> >
> > Furthermore, we recruited three professional human annotators to evaluate the quality ranking of images generated by IRG reasoning steps 1 and 2 on the WISE benchmark (specific details are provided in our response to W2 of Reviewer JwV8). The results are as follows:
> >
> > |                     | Qwen | GPT-4o | UnifiedReward | Human |
> > | ------------------- | ---- | ------ | ------------- | ----- |
> > | IRG reasoning step1 | 29%  | 38%    | 43%           | 17%   |
> > | IRG (ours)          | 71%  | 62%    | 57%           | 74%   |
> >
> > The human evaluation results demonstrate strong consistency with MLLM judge results. Human judges exhibit an even stronger preference for images generated at IRG reasoning step 2, which validates that IRG can significantly enhance image generation quality through multi-turn reasoning.
> >
> > Therefore, we believe that our proposed IRG makes significant contributions with substantial practical benefits, and is by no means characterized by "questionable contribution."
> >
> > > W2-2: Explanation of Failure Cases
> >
> > We appreciate your suggestion.
> >
> > We present and analyze several failure cases in Fig. 9 and Appendix D.6. As illustrated in Fig. 9, some cases attempt to rectify certain deficiencies in images generated at reasoning step 1. While these modifications succeed in improving specific details (green boxes), some aspects still fail to achieve the desired improvement (red boxes). Additionally, some cases attempt to modify the overall details of the image, but such modifications can be subjective—some human annotators still prefer images generated at reasoning step 1. Nevertheless, as demonstrated in Table 4, the improvements achieved by our proposed IRG are present in the majority of cases and better align with human preferences in terms of quality and detail.
> >
> > > General Q: Loss of Details in Refined Images in Fig. 4
> >
> > We must clarify that the refined images in Fig. 4 do not lose details due to degraded image quality. Rather, they optimize the overall composition and reduce visual noise from previously generated images. As shown in the figure, images from IRG reasoning step 2 significantly improve unclear, noisy, or erroneous generation components present in step 1, while placing greater emphasis on visual perception. To some extent, this refinement smooths certain visual noise, which may present an appearance similar to detail loss.

---

> > > ### Author Response · Authors · 2025-12-03
> > > **Rebuttal (3/3)**
> > >
> > > > Q1: Training and Data Construction Costs
> > >
> > > We appreciate your suggestion.
> > >
> > > Our IRG model was trained on four machines, each equipped with 8 H800 GPUs, for a total of approximately 40 hours. The construction of IRGL primarily consumed GPT-4o API resources, with approximately $6,000 in API costs for generating the image data.
> > >
> > > > Q2: Inference Cost
> > >
> > > Please refer to our response to W1.
> > >
> > > > Q3: Contradictory Evaluation Results
> > >
> > > Please refer to our response to W2-1 regarding the explanation of contradictory evaluation results. Additionally, please see our response to W2 of Reviewer JwV8 regarding human evaluation results, which demonstrate that IRG's second round of generation significantly improves image generation quality, consistent with MLLM judge results.
> > >
> > > > Reference
> > >
> > > [1] A Survey on LLM-as-a-Judge. EMNLP 2025.

---

### Official Review · Reviewer_5XBw · 2025-10-31

**Soundness:** 4
**Presentation:** 3
**Contribution:** 3
**Rating:** 8
**Confidence:** 3

**Summary:**

This paper proposes Interleaving Reasoning Generation (IRG), a framework that improve T2I generaton by interleaving reasoning.


The key idea is to let the model first "think" (generate textual reasoning), then generate an image conditioned on that reasoning, and finally "reflect" on the generated image to refine it.
The framework differs from previous "multi-turn post-hoc self-refinement / editing" methods, IRG integrates reasoning and refinement in a single end-to-end pipeline via the supervision of "text–thinking-image–thinking–image" trajectory.
To support this paradigm, this paper introduces IRGL-300K, a large-scale dataset with six decomposed learning modes designed to train both text-based reasoning and full image generation capabilities.


Comprehensive experiments demonstrate strong improvements (5–10 points) across multiple benchmarks (GenEval, WISE, TIIF, GenAI-Bench, OneIG-EN).


The approach is conceptually interesting, technically solid, and empirically effective.

**Strengths:**

1. The proposed Interleaving Reasoning Generation (IRG), which differs from previous  "multi-turn post-hoc self-refinement / editing" methods. Instead of iterative post-hoc image editing, IRG integrates reasoning and refinement in a single end-to-end pipeline via "text–thinking-image–thinking–image" supervision. This unified formulation is both elegant and innovative.

2. The approach is not limited to improving semantic alignment (as most prior reflection-based methods do) but also targets to improve visual quality, including texture rendering, fine-grained detail, and shadow realism.

3. The IRG pipeline achieves consistent gains of 5–10 points on five diverse text-to-image benchmarks (GenEval, WISE, TIIF, GenAI-Bench, and OneIG-EN), demonstrating strong generalization and scalability.

4. The authors recognize that standard benchmarks are often insensitive to fine-grained visual improvements and supplement them with multi-MLLM evaluator comparisons, providing convincing qualitative and quantitative validation.

**Weaknesses:**

Seen in Questions

**Questions:**

My questions focus on clarity and rationale.

1. Lines 288–290
> The main goal of this stage is to strengthen the text-based reasoning capability, while incorporating full thinking–image trajectories to avoid degrading the core generative performance.

It is unclear why "incorporating full thinking–image trajectories" is necessary during the reasoning-focused stage.
Was this motivated by empirical observation (e.g., observed degradation of image generation without such data)?
Or is it based on the general expectation that further fine-tuning of MLLMs may cause forgetting of generative capabilities?

2. Lines 199–200
> The model exchanges and exploits multiple segments of interleaved text–image representations, a process we term Interleaving Reasoning Generation (IRG).

The meaning of "exchanges and exploits multiple segments of interleaved text–image representations" is not clear.
Does this refer to multi-modal token interleaving within the transformer architecture,
or to multi-step reasoning across modalities (e.g., text → image → text → image)?
This statement could be misleading and might need clarification.

3. Section 2.2.2.

This section is quite dense and terminology-heavy.
It would be valuable to explain the thought process that led to the definition of the six decomposed learning modes.
Was this decomposition empirically derived or conceptually motivated?
A clear justification would significantly improve readability and insight.

Additionally, according to the mapping pattern (Eq. 4 →Eq.  7, Eq. 5 → Eq. 8, Eq. 6 → Eq. 9),
why does Eq. 7 not include T^{(1)}_{out} as an input, while Eq. 8 and Eq. 9 do?
This asymmetry deserves clarification.

4. Section 2.2.4.

You propose two complementary CFG-conditioning schemes (image-based and reflection-based).
Why not also include the traditional prompt-only CFG (conditioning on T_{in}) as a baseline?
Was this excluded for theoretical or empirical reasons (e.g., redundancy, instability, or negligible impact)?
Discussing this design choice would improve completeness.

5. Figure 2 is difficult to understand.

---

> ### Author Response · Authors · 2025-12-03
> **Rebuttal (1/1)**
>
> > W1: Rationale for Incorporating Full Thinking-Image Trajectories
>
> We sincerely appreciate your suggestion.
>
> The rationale for incorporating full thinking-image trajectories stems from the nature of our base model being a unified model, where generation and comprehension capabilities are tightly coupled. When training is limited to text-only supervision (CE loss), we observe significant degradation in generation performance. We conducted an ablation study removing image learning (*Initial Full Learning* and *Improving Full Learning*) during Stage 1 training, with results presented below:
>
> |                                           | WISE | TIIF        | GenAI-Bench |
> | ----------------------------------------- | ---- | ----------- | ----------- |
> | IRG w/o Image Learning in Stage1 Training | 0.56 | 55.82/54.31 | 0.72        |
> | IRG (ours)                                | 0.77 | 76.00/73.77 | 0.84        |
>
> This ablation study explicitly demonstrates the necessity of incorporating full thinking-image trajectories during Stage 1 training.
>
> > W2: Clarification of "Exchanges and Exploits Multiple Segments of Interleaved Text-Image Representations"
>
> We apologize for any ambiguity.
>
> The phrase "exchanges and exploits multiple segments of interleaved text-image representations" refers to "multi-step reasoning across modalities." We have revised the manuscript to provide a clearer articulation of this concept.
>
> > W3: Design Rationale for the Six Decomposed Learning Modes
>
> We appreciate your suggestion.
>
> The design of *Understanding Learning* (Eq. 4 and 7) and *Generation Learning* (Eq. 5 and 8) both aim to enable the model to learn text-based reasoning capabilities, with key distinctions in their approaches. *Understanding Learning* directly learns text-based reasoning grounded in visual information. For instance, *Initial Thinking Understanding Learning* learns to generate the initial thinking given an image, while *Improving Thinking Understanding Learning* learns to produce improving thinking given both the first and second round images.
>
> In contrast, *Generation Learning* is designed to directly simulate the actual IRG reasoning process. For example, *Initial Thinking Generation Learning* mimics the model's behavior of generating initial thinking directly from the given prompt, while *Improving Thinking Generation Learning* generates improving thinking based on the previously generated image and reasoning chain. This design is motivated by the intuition and empirical observation that learning thinking with visual reference versus learning thinking directly from text poses different learning challenges for the model—the former benefits from contrastive visual references, while the latter requires reasoning with more limited information.
>
> *Full Learning* (Eq. 6 and 9) extends *Generation Learning* by incorporating high-quality image generation learning (MSE loss) to enhance image generation quality.
>
> Regarding why Eq. 7 does not include $T^{(1)}_{out}$ as input: we designed a specific question $T_q$ that instructs the model to learn the improvements from the initial image to the improved image during *Improving Thinking Understanding Learning*. The actual question used is as follows:
>
> ```
> You have been given one prompt and two images. You should carefully compare
> the first generated image with the second one, identify specific visual
> differences and improvements, and generate a detailed "think process" that
> explains how to transform the previous image into the improved version.
>
> The improvement think process must follow this structure:
> 1. Start with <think> and end with </think>
> 2. Have clear paragraph breaks with the following sections:
>    - Initial analysis paragraph (acknowledge shortcomings of the previous image)
>    - ### Detailed Explanation of Required Improvements:
>    - ### Step-by-Step Modification Guidance:
>    - ### Final Comprehensive Prompt for the Improved Image:
>
> Each section should provide concrete, actionable guidance for text-to-image
> generation, focusing on composition, lighting, colors, textures, positioning,
> and style. Ensuring the think process is specific enough to guide precise
> image improvements.
> ```
>
> > W4: CFG Strategy
>
> We provide comprehensive comparisons and explanations of CFG strategies in Fig. 5 and Appendix D.2. Specifically, "w/ naive CFG" in Fig. 5 corresponds to the approach you mentioned—applying traditional prompt-only CFG at reasoning step 2. As demonstrated, naive CFG still leads to unstable generation results. Through our validation, the proposed CFG strategy consistently produces more stable image quality across multiple generation steps.
>
> We have further strengthened the descriptions in the caption of Fig. 5 and in Appendix D.2 for clarity.
>
> > W5: Improving Clarity of Figure 2
>
> We appreciate your suggestion. We have enhanced the caption of Fig. 2 with more detailed descriptions to improve comprehensibility and reduce potential confusion.

---

### Official Review · Reviewer_JwV8 · 2025-11-04

**Soundness:** 3
**Presentation:** 2
**Contribution:** 3
**Rating:** 6
**Confidence:** 4

**Summary:**

The paper introduced Interleaving Reasoning Generation (IRG). The main idea is to reason with both text and image during text-to-image, to produce better images in the end. Specifically, the process in this paper has 4 stages: (1) Given an input text prompt, generate textural "thinking" (2) after thinking, generate an initial image (3) generate the second textural "thinking" to refine the image (4) generate the final image based on the previous steps.

The training process includes 6 learning modes, each targeting different sub-steps above. The authors also created IRGL-300K, a training dataset for this kind of interleaved reasoning generation. The primary source of the images are from GPT-4o. The authors also used Qwen2.5-VL to generate the text data.

The authors conduct experiment on Bagel, showing that it improves the results on GenEval, WISE, TIIF, and GenAI-Bench. The authors also conduct ablation studies on various components in the training process.

**Strengths:**

1. The idea of let unified models to use interleaved text and image reasoning steps for text-to-image generation makes a lot of sense. It utilizes unified models' power, and goes beyond bagel's COT + image generation scheme.

2. Experiments show that the authors' approach works, and can give improvement compared to the original Bagel.

3. The authors conduct experiments on a variety of tasks, and the ablations answer many important research questions.

**Weaknesses:**

1. The method is heavily dependent on distilling GPT-4o images. The dataset for "initial learning stage" is adapted from a GPT-4o generated dataset. The second stage is created through GPT-4o and Bagel generated images. It remains an open question that if this method works without a strong text-to-image model to distill with. For example, if the authors can observe similar gains through data generated by another open text-to-image model? Or even from Bagel's own generations? That would make the paper much stronger.

2. There is no human evaluation in this paper. All evaluations are model based, and some are MLLM-as-a-judge results. It would be benefitial to show some human results on a subset of the testing prompts, to confirm that this method indeed improves model performance.

3. There is no discussion on the compute / inference cost of this method. With more stages and many CFG steps, there will be a lot of extra compute, and they worth discussion.

4. It would be beneficial to show some more analysis on the final model. For example, what is the lengths of the thinking steps? Are there any failure examples that the second image is not improving the first image?

**Questions:**

Most of my questions have been discussed in the weakness section

1. Can this method work without using GPT-4o distilled data? For example, using Qwen-Image. How will the performance change.

2. Is it possible that with Bagel's self-generated data, and some real images, we can see such gains?

3. More analysis on the reasoning steps would be very helpful. Like more analysis on the reasoning lengths, and some failure case examples.

4. What is the inference cost of the method? Is it possible that in some cases, we can just finish generation in the first stage, to save some cost.

---

> ### Author Response · Authors · 2025-12-03
> **Rebuttal (1/2)**
>
> > W1: Dataset Construction with Open-Source Models
>
> We sincerely appreciate your suggestion.
>
> Regarding the feasibility of utilizing open-source models for constructing our IRGL dataset, we believe this approach is indeed viable. As illustrated in Fig. 7, we provide several cases comparing images generated by open-source models and our adopted GPT-4o model. The results demonstrate that both Hunyuan-Image-3.0 [1] and GPT-4o achieve significant and correct improvements based on the initial images generated by BAGEL. The core principle of our IRGL dataset construction is that the improved images should exhibit higher quality in terms of alignment with the original prompt and demonstrate clear improvements over the initial images. Our analysis shows that the open-source Hunyuan-Image-3.0 successfully accomplishes this objective, indicating that constructing datasets with open-source models to eliminate dependence on the expensive GPT-4o API is completely feasible.
>
> Furthermore, we observe that Qwen-Image-Edit shows limited improvement over the initial images. We hypothesize that this is because Qwen-Image-Edit is not a unified model and has relatively fewer parameters, making it challenging to meet our requirements for generating improvements that better align with the original prompt while building upon the initial image. In contrast, Hunyuan-Image-3.0 and GPT-4o, benefiting from their unified architecture, stronger comprehension capabilities, and substantially larger parameter counts, are better equipped to achieve our objectives.
>
> > W2: Human Evaluation
>
> We greatly appreciate your valuable suggestion.
>
> We recruited three professional human annotators to evaluate the image quality generated by IRG at reasoning steps 1 and 2 on the WISE benchmark. The annotators were instructed to compare the quality of two images based on the original prompt, with the order of images from the two IRG rounds randomly shuffled. They were allowed to select from three options: "A is better," "B is better," or "Equally good." The aggregated results are presented below:
>
> |                     | Qwen | GPT-4o | UnifiedReward | Human |
> | ------------------- | ---- | ------ | ------------- | ----- |
> | IRG reasoning step1 | 29%  | 38%    | 43%           | 17%   |
> | IRG (ours)          | 71%  | 62%    | 57%           | 74%   |
>
> The human evaluation results demonstrate strong consistency with the MLLM judge results. Notably, human judges exhibit an even stronger preference for images generated at IRG reasoning step 2, which validates that IRG can significantly enhance image generation quality through multi-turn reasoning.
>
> Additionally, we have incorporated visualization results of the human evaluation in Fig. 6.
>
> > W3: Inference Cost
>
> We appreciate your suggestion and have supplemented information regarding inference efficiency.
>
> We measured the inference latency and peak GPU memory consumption on a single H800 GPU, averaged over 100 prompts. The results are as follows:
>
> |                    | IRG reasoning step 1 | IRG reasoning step 2 |
> | ------------------ | -------------------- | -------------------- |
> | Inference latency  | 29.79s               | 60.58s               |
> | Peak GPU memory    | 28.23GB              | 29.35GB              |
>
> As shown in the table, the two-step reasoning process of IRG does introduce additional inference overhead. However, this overhead primarily manifests in inference latency rather than peak GPU memory consumption (where the majority of GPU memory is occupied by model parameter loading). In this scenario, deploying this model for interleaved reasoning generation in practical applications is generally not constrained, as it does not require substantially additional GPU memory that would limit device deployment. The primary impact is on generation latency, which aligns with the bottleneck observed in LLM reasoning models.
>
> Furthermore, we would like to clarify that this additional overhead is an inherent characteristic of test-time scaling techniques. When we aim to enhance model performance through inference-time techniques, such overhead is expected and unavoidable compared to non-reasoning models, and is not a limitation unique to our approach. Our proposed IRG introduces both text-based and visual-based reasoning into the text-to-image generation domain, representing an innovative contribution to this field and providing a powerful new paradigm for further enhancing generative model capabilities.

---

> > ### Author Response · Authors · 2025-12-03
> > **Rebuttal (2/2)**
> >
> > > W4: Reasoning Length and Failure Cases
> >
> > We thank you for your suggestion.
> >
> > We have computed the average word count of IRG reasoning on the benchmark, as shown in the table below:
> >
> > |                     | Word count |
> > | ------------------- | ---------- |
> > | IRG reasoning step1 | 243.45     |
> > | IRG reasoning step2 | 582.55     |
> >
> > The results indicate that the text-based reasoning process in IRG reasoning step 1 produces relatively shorter chain-of-thought (CoT) compared to IRG reasoning step 2. This is because we designed a more quality-focused, multi-perspective template CoT for the improving thinking process in IRG reasoning step 2 (complete CoT shown in Figs. 11 and 12), which naturally results in longer reasoning chains.
> >
> > Additionally, we present and analyze several failure cases in Fig. 9 and Appendix D.6. As illustrated in Fig. 9, some cases attempt to rectify certain deficiencies in the images generated at reasoning step 1. While these modifications succeed in improving specific details (green boxes), some aspects still fail to achieve the desired improvement (red boxes). Moreover, some cases attempt to modify the overall details of the image, but such modifications can be subjective—some human annotators still prefer the images generated at reasoning step 1. Nevertheless, as demonstrated in Table 4, the improvements achieved by our proposed IRG are present in the majority of cases and better align with human preferences in terms of quality and detail.
> >
> > > Q1: Alternative to GPT-4o
> >
> > Please refer to our response to W1. Using open-source models such as Hunyuan-Image-3.0 instead of GPT-4o is completely feasible. However, for smaller-parameter, non-unified generative models like Qwen-Image-Edit, challenges in data quality persist.
> >
> > > Q2: Dataset Construction Using BAGEL Itself or Real Images
> >
> > We would like to clarify that all initial images in our dataset are generated by BAGEL itself. Our objective is to enable the model to learn the steps and details for improving toward higher-quality images based on its own generated outputs. This learning process relies on a relatively powerful generative model to guide BAGEL's improvement. If we were to use BAGEL to improve its own initial images, the quality of the improved images might be insufficient, making it difficult to enhance the model's final generation quality.
> >
> > Regarding the use of real images, we believe this is theoretically feasible (since the core of our IRGL full reasoning trajectory construction focuses on the quality of improved images). However, in practice, this approach faces challenges in pairing improved images with initial images. As discussed in lines 296-309: "IRG requires paired samples linking an initial image to its improved counterpart. Designing the transformation process from the initial image to the improved image is itself a challenge." While using high-quality real-world images is feasible, it may result in substantial variations between the initial and improved images, making such learning potentially difficult.
> >
> > > Q3: Reasoning Length and Failure Cases
> >
> > Please refer to our response to W4.
> >
> > > Q4: Inference Cost and Early Termination
> >
> > For IRG inference costs, please refer to our response to W3.
> >
> > Regarding whether generation can be completed at the first stage, we conducted an experimental study on the WISE benchmark. We employed the Qwen3-VL [2] model to judge whether the IRG reasoning process can be terminated early, given the prompt. The results are as follows:
> >
> > |                     | Early stop rate | WISE |
> > | ------------------- | --------------- | ---- |
> > | IRG reasoning step1 | -               | 0.79 |
> > | IRG (ours)          | -               | 0.77 |
> > | IRG w/ early stop   | 71%             | 0.79 |
> >
> > As shown in the table, introducing an MLLM to judge early termination is feasible (at least without degrading benchmark metrics). However, this introduces a fundamental issue: the MLLM lacks reference information, making it difficult to determine whether subsequent IRG steps will yield further improvements, potentially leading to erroneous judgments.
> >
> > As illustrated in Fig. 8, we present visualization results of MLLM judgments. In some cases, the MLLM judge prematurely terminates the IRG reasoning process, while the IRG-improved images actually demonstrate superior quality compared to the initial images. This represents a potential limitation of this approach. We believe this constitutes a promising direction for future work.
> >
> > > References
> >
> > [1] HunyuanImage 3.0 Technical Report. arXiv:2509.
> >
> > [2] Qwen3-VL Technical Report. arXiv:2511.

---

### Author Response · Authors · 2025-12-03
**Revision Summary**

Dear Reviewers,

We thank all reviewers for your insightful and constructive comments! We have submitted the point-by-point responses to all reviewers' comments, along with a revised version of the paper **(with blue text)**.

We thank ***all reviewers for recognizing the contribution, novelty, and effectiveness of our IRG method***. Below, we provide a summary of the specific revisions made to the paper.

Sincerely Yours,

The Authors

- [Reviewer PpTp] Increase the readability in Eq. 1 (Eq. 1)
- [Reviewer 5XBw] Increase the more explanation for Fig. 2 (Fig.2 captions)
- [Reviewer JwV8, bEWK] Human judge for IRG results (Tab. 4, Sec. 3.2)
- [Reviewer PpTp] More discussion of related works (Appendix A.2)
- [Reviewer 5XBw] Clarification of CFG strategy (Fig. 5 captions)
- [Reviewer JwV8] Early stop experiments for IRG (Tab. 8, Fig. 8, Appendix D.5)
- [Reviewer bEWK] More analysis for relation in benchmark score and image quality. (Fig. 6, Appendix D.3)
- [Reviewer JwV8] Discussion of open-source model for dataset conduction (Fig. 7, Appendix D.4)
- [Reviewer 5XBw, PpTp] More detailed ablation study of IRG (Tab. 10, Appendix D.8)
- [Reviewer 5XBw] More analysis for the motivation of Decomposed Learning Modes (Appendix D.8)
- [Reviewer JwV8, bEWK] More analysis of failure cases (Fig. 9, Appendix D.6)
- [Reviewer JwV8, bEWK, PpTp] Analysis of IRG inference cost (Tab. 11, Appendix D.9)
- [Reviewer PpTp] Discussion for extended reasoning steps (Tab. 9, Fig. 10, Appendix D.7)
- [Reviewer PpTp] More intermediate reasoning visualizations (Fig. 11, 12, Appendix D.10)

---

### Meta-Review · Area_Chair_vhQQ · 2026-01-06

**Summary:**

The major concerns raised by the reviewers include
1. Heavy dependency on GPT-4o and unclear generalizability
2. Lack of human evaluation and inference cost discussion
3. Unclear practical value of the complex approach
4. In sufficient comparison with related works

**Reviewer Concerns:**

1. The authors provide additional observations to address the generalizability problem. Although this does not fully resolve the concern, it is a common limitation in related work and is not a major problem.
2. The authors include human evaluation and inference cost analysis in the rebuttal, thereby addressing the corresponding concerns.
3. The arguments supporting the value and practicality of the proposed approach are not convincing and do not fully resolve the concern; it remains unclear how valuable the approach is in real-world applications.
4. While the authors discuss differences from related work, the discussion does not fully clarify the contribution of this work, and direct comparisons remain desirable.

**Reviewer Scores:**

For reviewers with initially positive ratings, their major concerns are likely to be addressed through the rebuttal and further discussion, and they are likely to maintain or raise their ratings. For reviewer PpTp, the concerns may also be resolved during the discussion phase; in particular, the reviewer indicated a willingness to raise the rating if additional results were provided, which has been addressed in the rebuttal. For reviewer bEWK, it is less clear whether the rebuttal and discussion will adequately address their concerns, particularly given the broader scope of the questions raised. Overall, the paper is likely to benefit from further discussion.

---

### Decision · Program_Chairs · 2026-01-26

Accept (Poster)